# Scalable Private Partition Selection via Adaptive Weighting

**Justin Y. Chen** [1]  **Vincent Cohen-Addad** [2]  **Alessandro Epasto** [2]  **Morteza Zadimoghaddam** [3]

## Abstract

In the differentially private partition selection problem (a.k.a. private set union, private key discovery), users hold subsets of items from an unbounded universe. The goal is to output as many items as possible from the union of the users' sets while maintaining user-level differential privacy. Solutions to this problem are a core building block for many privacy-preserving ML applications including vocabulary extraction in a private corpus, computing statistics over categorical data and learning embeddings over user-provided items. We propose an algorithm for this problem, `MaxAdaptiveDegree` (`MAD`), which adaptively reroutes weight from items with weight far above the threshold needed for privacy to items with smaller weight, thereby increasing the probability that less frequent items are output. Our algorithm can be efficiently implemented in massively parallel computation systems allowing scalability to very large datasets. We prove that our algorithm stochastically dominates the standard parallel algorithm for this problem. We also develop a two-round version of our algorithm, `MAD2R`, where results of the computation in the first round are used to bias the weighting in the second round to maximize the number of items output. In experiments, our algorithms provide the best results among parallel algorithms and scale to datasets with hundreds of billions of items, up to three orders of magnitude larger than those analyzed by prior sequential algorithms.

## 1. Introduction

The availability of large amounts of user data has been one of the driving factors for the widespread adoption and rapid development of modern machine learning and data analytics.

Consider the example of a system releasing information on the queries asked to a search engine over a period of time (Korolova et al., 2009; Bavadekar et al., 2021). Such a system can provide valuable insights to researchers and the public (for instance on health concerns Bavadekar et al. (2021)) but care is needed in ensuring that the queries output do not leak private and sensitive user information.

In this paper, we focus on the problem of *private partition selection* (Desfontaines et al., 2022; Gopi et al., 2020) which models the challenge of extracting as much data as possible from such a dataset while respecting user privacy. More formally, the setting of the problem (which is also known as private set union or private key discovery) is that each user has a private subset of items (e.g., the queries issued by the user) from an unknown and unbounded universe of items (e.g., all strings). The goal is to output as many of the items in the users' sets as possible (i.e., the queries issues by the users), while providing a strong notion of privacy—User-level Differential Privacy (DP) (Dwork & Roth, 2014).

Private partition selection models many challenges beyond the example above, including the problem of extracting the vocabulary (words, tokens or $n$-gram) present in a private corpus (Zhang et al., 2022; Kim et al., 2021). This task is a fundamental prerequisite for many privacy-preserving natural language processing algorithms (Wilson et al., 2019; Gopi et al., 2020), including for training language models for sentence completion and response generation for emails (Kim et al., 2021). Similarly, learning embedding models over categorical data often requires to identify the categories present in a private dataset (Ghazi et al., 2023; 2024). Partition selection underpins many other applications including analyzing private streams (Cardoso & Rogers, 2022; Zhang et al., 2023), learning sparse histograms (Boneh et al., 2021), answering SQL queries (Desfontaines et al., 2022) and sparsifying the gradients in the DP SGD method (Ghazi et al., 2023). Unsurprisingly given these applications, private partition selection algorithms (Korolova et al., 2009) are a core building block of many standard differentially private libraries e.g., PyDP (PyDP, 2024), Google's DP Libraries (Google, 2024; Amin et al., 2022), and OpenMined DP Library (OpenMined, 2024).

Real-world datasets for these applications can be massive, potentially containing hundreds of billions of data

---
[1]Massachusetts Institute of Technology, Cambridge, MA, USA [2]Google Research, New York, NY, USA [3]Google Research, Zurich, Switzerland. Correspondence to: Justin Chen <justc@mit.edu>.

*Proceedings of the 42nd International Conference on Machine Learning*, Vancouver, Canada. PMLR 267, 2025. Copyright 2025 by the author(s).

points, thus requiring algorithms for partition selection that can be efficiently run in large-scale data processing infrastructures—e.g., MapReduce (Dean & Ghemawat, 2004), Hadoop (Apache Software Foundation), Spark (Zaharia et al., 2016). In our work, we design a highly parallelizable algorithm for this problem which requires constant parallel rounds in the Massively Parallel Computing model (Karloff et al., 2010) and does not assume to fit the input in memory. This contrasts to prior algorithms such as (Gopi et al., 2020; Carvalho et al., 2022) which all require (with the key exception of the uniform weighting method described below) to process all the data sequentially on a single machine and assume storing the input in-memory thus precluding efficient parallelization.

## 1.1. Weight and Threshold Approach

Before introducing our algorithm, we review the popular weighting-based approach to partition selection which is used in many algorithms (Korolova et al., 2009; Gopi et al., 2020; Carvalho et al., 2022; Swanberg et al., 2023). This approach is of interest in the context of large-scale data as some of its variants can be parallelized efficiently (Korolova et al., 2009; Swanberg et al., 2023).

Notice that differential privacy imposes to not output any item which is owned by only a single user. However, it is possible for a private algorithm to output items which appear in *many* different sets. This intuition is at the basis of the weighting-based algorithms.

Algorithms in this framework start by subsampling each user's set to bound the maximum number of items per user. Then, these algorithms proceed by increasing, for each user, the weight associated the items present in the user data. Finally, the algorithm adds Gaussian (or Laplace) noise to the total accumulated weight of each item, and outputting all items with noised weight above a certain threshold (Korolova et al., 2009; Gopi et al., 2020; Carvalho et al., 2022; Swanberg et al., 2023). The amount of noise and value of the threshold depends on the privacy parameters and *crucially* on the sensitivity of the weighting function to the addition or removal of any individual user's set. Loosely speaking, the contribution of each user to the item weights must be bounded in order to achieve differential privacy. Algorithms within this framework differ in the choice of how to assign item weights, but in all designs the key goal is that of limiting the sensitivity of the weighting function.

A basic strategy is *uniform weighting* (Korolova et al., 2009) where each user contributes equal weight to each of the items in their set. It is easy to bound the sensitivity of this basic weighting and thus to prove differential privacy. Because of its simplicity, the basic uniform weighting algorithm is extremely parallelizable requiring only basic counting operations over the items in the data.

Unfortunately, however, uniform weighting is lossy in that it may overallocate weight far above the threshold to high frequency items, missing an opportunity to boost the weight of items closer to the decision boundary. This has inspired the design of greedy weighting schemes such as (Gopi et al., 2020; Carvalho et al., 2022) where each user's allocations depend on data of previously analyzed users. All of these algorithms are inherently sequential and require memory proportional to the items present in the data.

To our knowledge, the uniform weighting (Korolova et al., 2009) is essentially the only known solution to the private partition selection problem which is amenable to implementation in a massively parallel computation framework. The sole exception is the scalable, iterative partition selection (DP-SIPS) scheme of (Swanberg et al., 2023) which has as core computation repeated invocations of the uniform weighting algorithm.

## 1.2. Our Contributions

In this work, we design the first, adaptive, non-uniform weighting algorithm that is amenable to massively parellel implementations. Our algorithm, called `MaxAdaptiveDegree` (MAD), requires linear work in the size of the input and can be implemented in a constant number of rounds in a parallel framework. From a technical point of view, the algorithm is based on a careful rerouting of overallocated weight to less frequent items, that together with a delicate sensitivity analysis shows no privacy loss compared to uniform weighting. This means that—given the same privacy parameters—both algorithms utilize exactly the same amount of noise and the same threshold (but our algorithm can better allocate the weight). As a result, we are able to prove that our algorithm stochastically dominates the basic, uniform weighting strategy.

We extend our result to multiple rounds in `MaxAdaptiveDegreeTwoRounds` (MAD2R), splitting our privacy budget across the rounds, running MAD in each round, and outputting the union of items found in both rounds. Similar to DP-SIPS, in the second round, we remove from the input any items found in the first round (this is private by post-processing). By a careful generalization of the privacy analysis of the weight and threshold approach, we show that it is possible to also use the noisy weights from the prior round. We leverage this in two ways. First, we additionally remove items which have very small weights from the first round–these have little chance of being output in the second round. Second, we bias the weighting produced by MAD in the second round to further limit overallocation to items which received large weights in the first round. The combination of these ideas yields significant empirical improvements over both the basic algorithm and DP-SIPS.

In MAD, users with a too small or large of a cardinality (we equivalently refer to this as the user's degree) are labeled non-adaptive: these users will add uniform weight to their items (or biased weight in the case of MAD2R). The rest of the users participate in adaptive reweighting with the privacy analysis making use of the upper bound on their degree. Initially each of these users sends a small amount of weight uniformly among their items (the total amount of weight sent per user is bounded by 1 rather than the square root of their degree, which is the case for the basic weighting algorithm). Then, items with weight significantly above the threshold are truncated to only have weight slightly above the threshold (we do not want to truncate all the way to the threshold as the added noise can decrease weights). The weight removed via truncation is returned to each user proportional to their initial contributions. Then, users reroute a carefully chosen fraction of this "excess" weight back to their items. Finally, users add additional uniform weight to their items to make up for the small amount of weight that was initially sent.

Bounding the sensitivity of MAD requires a careful analysis and is significantly more involved than for basic weighting. Several design choices made in our algorithm, such as using an initial uniform weighting inversely proportional to cardinality rather than square root of cardinality, using a minimum and maximum adaptive degree, and choosing the fraction of how much excess weight to reroute are all required for the following theorem to hold. Furthermore, we generalize the analysis of the weight and threshold paradigm to allow us to use noisy weights from first round in biasing weights in the second round of MAD2R. This biasing further complicates the sensitivity analysis of MAD which we address by putting limits on the minimum and maximum bias.

**Theorem 1.1** (Privacy, Informal version of Theorem B.1 and Corollary B.2). *Using MAD as the weighting algorithm achieves $(\varepsilon, \delta)$-DP with the exact same noise and threshold parameters as the basic algorithm. Running MAD in two rounds with biases via MAD2R is $(\varepsilon, \delta)$-DP.*

Within MAD, items have their weight truncated if it exceeds an "adaptive threshold" $\tau$ after adding the initial weights. $\tau$ is set to be $\beta$ standard deviations of the noise above the true threshold that will be used to determine the output where $\beta \geq 0$ is a free parameter of the algorithm. By design, before adding noise, every item which receives at least weight $\tau$ in the basic algorithm will also receive weight at least $\tau$ by MAD. Furthermore, the weights on all other items will only be increased under MAD compared to the basic algorithm. Taking the final step of adding noise, we show the following theorem.

**Theorem 1.2** (Stochastic Dominance, Informal version of Theorem C.1). *Let $U$ be the set of items output when using*

*the basic algorithm and let $U^*$ be the set of items output when using MAD as the weighting algorithm. Then, for items $i \in \mathcal{U}$ and a free parameter $\beta \geq 0$,*

- *If $\Pr(i \in U) < \Phi(\beta)$, then $\Pr(i \in U^*) \geq \Pr(i \in U)$.*
- *Otherwise, $\Pr(i \in U^*) \geq \Phi(\beta)$.*

*where $\Phi$ is the standard Gaussian cdf.*

Compared to the basic algorithm, MAD has a higher probability of outputting any item that does not reach the adaptive threshold in its initial stage as it reroutes excess weight to these items. The theorem shows that MAD *stochastically dominates* the basic algorithm on these items. For the remaining items, they already have an overwhelming probability of being output as their final weight before adding noise is at least several standard deviations above the threshold (this is quantitatively controlled by the parameter $\beta$). In Appendix C, we also describe a simple, concrete family of instances where MAD significantly improves upon the baselines.

Finally, we conduct experiments on several publicly-available datasets with up to 800 billions of (user, item) pairs (up to three orders of magnitude larger than prior datasets used in sequential algorithms). Our algorithm outperforms scalable baselines and is competitive with the sequential baselines.

### 1.3. Related Work

Our algorithms are in the area of privacy preserving algorithms with differential privacy guarantee which is the de facto standard of privacy (we refer to (Dwork & Roth, 2014) for an introduction to this area). As we covered the application and prior work on private partition selection in the introduction, we now provide more details on the work most related to our paper.

The differentially private partition selection problem was first studied in (Korolova et al., 2009). They utilized the now-standard approach of subsampling to limit the number of items in each user's set, constructing weights over items, and thresholding noised weights to produce the output. They proposed a version of the basic weighting algorithm which uses the Laplace mechanism rather than the Gaussian mechanism. This algorithm was also used in (Wilson et al., 2020) within the context of a private SQL system. The problem received renewed study in (Gopi et al., 2020) where the authors propose a generic class of greedy, sequential weighting algorithms which empirically outperform basic weighting (with either the Laplace or Gaussian mechanism). (Carvalho et al., 2022) gave an alternative greedy, sequential weighting algorithm which leverages item frequencies in cases where each user has a multiset of items. (Desfontaines et al., 2022)

analyzed in depth the optimal strategy when each user has only a single item (all sets have cardinality one). This is the only work that does not utilize the weight and threshold approach, but it is tailored only for this special case. The work most related to ours is DP-SIPS (Swanberg et al., 2023) which proposes the first algorithm other than basic weighting which is amenable to implementation in a parallel environment. DP-SIPS splits the privacy budget over a small number of rounds, runs the basic algorithm as a black box each round, and iteratively removes the items found in previous rounds for future computations. This simple idea leads to large empirical improvements, giving a scalable algorithm that has competitive performance with sequential algorithms.

## 2. Preliminaries

**Definition 2.1** (Differentially-Private Partition Selection). In the differentially-private partition selection (a.k.a. private set union or key selection) problem, there are $n$ users with each user $u$ having a set $S_u$ of items from an unknown and possibly infinite universe $\Sigma$ of items: the input is of the form $\mathcal{S} = \{(u, S_u)\}_{u \in [n]}$. The goal is to output a set of items $U$ of maximum cardinality, such that $U$ is a subset of the union of the users' sets $\mathcal{U} = \cup_{u \in [n]} S_u$, while maintaining user-level differentially privacy.

As standard in prior work (Korolova et al., 2009; Gopi et al., 2020; Carvalho et al., 2022; Swanberg et al., 2023) we consider the central differential privacy model, where the input data is available to a curator that runs the algorithm and wants to ensure differential privacy for the output of the algorithm. We now formally define these notions.

**Definition 2.2** (Neighboring Datasets). We say that two input datasets $\mathcal{S}$ and $\mathcal{S}'$ are neighboring if one can be obtained by removing a single user's set from the other, i.e., $\mathcal{S}' = \mathcal{S} \cup \{(v, S_v)\}$ for some new user $v$.

**Definition 2.3** (Differential Privacy (Dwork & Roth, 2014)). A randomized algorithm $\mathcal{M}$ is $(\varepsilon, \delta)$-differentially private, or $(\varepsilon, \delta)$-DP, if for any two neighboring datasets $\mathcal{S}$ and $\mathcal{S}'$ and for any possible subset of outputs $\mathcal{O} \subseteq \{U : U \subseteq \Sigma\}$,

$$\Pr(\mathcal{M}(\mathcal{S}) \in \mathcal{O}) \leq e^\varepsilon \cdot \Pr(\mathcal{M}(\mathcal{S}') \in \mathcal{O}) + \delta.$$

Let $\Phi : \mathbb{R} \to \mathbb{R}$ be the standard Gaussian cumulative density function.

**Proposition 2.4** (Gaussian Mechanism (Balle & Wang, 2018)). *Let $f : \mathcal{D} \to \mathbb{R}^d$ be a function with $\ell_2$ sensitivity $\Delta_2$. For any $\varepsilon > 0$ and $\delta \in (0, 1]$, the mechanism $M(x) = f(x) + Z$ with $Z \sim \mathcal{N}(0, \sigma^2 I)$ is $(\varepsilon, \delta)$-DP if*

$$\Phi\left(\frac{\Delta_2}{2\sigma} - \frac{\varepsilon\sigma}{\Delta_2}\right) - e^\varepsilon \Phi\left(-\frac{\Delta_2}{2\sigma} - \frac{\varepsilon\sigma}{\Delta_2}\right) \leq \delta.$$

---

**Algorithm 1** Meta-algorithm for private partition selection. WeightAndThreshold($\mathcal{S}, \varepsilon, \delta, \Delta_0, \text{ALG}, h$)

**Input:** User sets $\mathcal{S} = \{(u, S_u)\}_{u \in [n]}$, privacy parameters $(\varepsilon, \delta)$, degree cap $\Delta_0$, weighting algorithm ALG, upper bound on the novel $\ell_\infty$ sensitivity, function $h : \mathbb{N} \to \mathbb{R}$

**Output:** Subset of the union of user sets $U \subseteq \mathcal{U} = \cup_{u=1}^n S_u$, noisy weight vector $\tilde{w}_{ext}$

1: Select $\sigma$ corresponding to the Gaussian Mechanism (Proposition 2.4) for $(\varepsilon, \delta/2)$-DP with $\Delta_2 = 1$.

2: Set $\rho \leftarrow \max_{t \in [\Delta_0]} h(t) + \sigma \Phi^{-1}\left(\left(1 - \frac{\delta}{2}\right)^{1/t}\right)$

3: **for all** $u \in [n]$ **do**
4:    **if** $|S_u| \geq \Delta_0$ **then**
5:       Randomly subsample $S_u$ to $\Delta_0$ items.    ▷ Cap user degrees.
6:    **end if**
7: **end for**
8: $w \leftarrow \text{ALG}(\mathcal{S})$    ▷ Weights on items in $\mathcal{U}$
9: $\tilde{w}(i) \leftarrow w(i) + \mathcal{N}(0, \sigma^2 I)$    ▷ Add noise
10: $U \leftarrow \{i \in \mathcal{U} : \tilde{w}(i) \geq \rho\}$    ▷ Apply threshold.

11: $\tilde{w}_{ext}(i) \leftarrow \begin{cases} \tilde{w}(i) & \text{if } i \in \mathcal{U} \\ \mathcal{N}(0, \sigma^2 I) & \text{if } i \in \Sigma \setminus \mathcal{U} \end{cases}$

   ▷ The $i \in \Sigma \setminus \mathcal{U}$ part is only for privacy analysis (we only ever query this vector on $i \in \mathcal{U}$).

12: **return** $U, \tilde{w}_{ext}$

---

## 3. Weight and Threshold Meta-Algorithm

In this section, we formalize the weighting-based meta-algorithm used in prior solutions to the differentially private partition selection problem (Korolova et al., 2009; Gopi et al., 2020; Carvalho et al., 2022; Swanberg et al., 2023). Our algorithm MAD also falls within this high-level approach with a novel weighting algorithm that is both adaptive and scalable to massive data. We alter the presentation of the algorithm from prior work in a subtle, but important, way by having the algorithm release a noisy weight vector $\tilde{w}_{ext}$ in addition to the normal set of items $U$. This allows us to develop a two-round version of our algorithm MAD2R which queries noisy weights from the first round to give improved performance in the second round, leading to signficant empirical benefit.

The weight and threshold meta-algorithm is given in Algorithm 1. Input is a set of user sets $\mathcal{S} = \{(u, S_u)\}_{u \in [n]}$, privacy parameters $\varepsilon$ and $\delta$, a maximum degree cap $\Delta_0$, and a weighting algorithm ALG (which can itself take some optional input parameters), and a function $h : \mathbb{N} \to \mathbb{R}$ which describes the sensitivity of ALG.

First, each user's set is randomly subsampled so that the size of each resulting set is at most $\Delta_0$ (the necessity of this step will be further explicated). Then, the ALG takes in the cardinality-capped sets and produces a set of weights over all items in the union. Independent Gaussian noise with standard deviation $\sigma$ is added to each coordinate of the

weights, and items with weight above a certain threshold $\rho$ are output. By construction, this algorithm will only ever output items which belong to the true union, $U \subseteq \mathcal{U}$, with the size of the output depending on the number of items with noised weight above the threshold.

For the sake of analysis (and not the implementation of the algorithm), we diverge from prior work to return a vector $\tilde{w}_{ext}$ of noisy weights over the entire universe $\Sigma$. This vector is implicitly used in the proof of privacy for releasing the set of items $U$, but it is never materialized as $|\Sigma|$ is unbounded. Within our algorithms, we will ensure that we only ever query entries of this vector which belong to $\mathcal{U}$, so we only ever have to materialize those entries. Note, however, that it would *not* be private to release $\tilde{w}$ as the output of a final algorithm as the domain of that vector is exactly the true union of the users' sets.

The privacy of this algorithm depends on certain "sensitivity" properties of ALG as well as our choice of $\sigma$ and $\rho$. Consider any pair of neighboring inputs $\mathcal{S}$ and $\mathcal{S}' = \mathcal{S} \cup \{(v, S_v)\}$, let $\mathcal{U}$ and $\mathcal{U}'$ be the corresponding unions, and let $w$ and $w'$ be the item weights assigned by ALG on the two inputs, respectively.

**Definition 3.1.** The $\ell_2$ *sensitivity* of a weighting algorithm is defined as the smallest value $\Delta_2$ such that,

$$\Delta_2 \geq \sqrt{\sum_{i \in \mathcal{U}} \left(w'(i) - w(i)\right)^2 + \sum_{i \in \mathcal{U}' \setminus \mathcal{U}} w'(i)^2}.$$

Given bounded $\ell_2$ sensitivity, choosing the scale of noise $\sigma$ appropriately for the Gaussian mechanism in Proposition 2.4 ensures that outputting the noised weights on items in $\mathcal{U}$ satisfies $\left(\varepsilon, \frac{\delta}{2}\right)$-DP. So if we knew $\mathcal{U}$, then the output of the algorithm after thresholding would be private via post-processing.

However, knowledge of the union $\mathcal{U}$ is exactly the problem we want to solve. The challenge is that there may be items in $\mathcal{U}'$ which do not appear in $\mathcal{U}$. Let $T = \mathcal{U}' \setminus \mathcal{U}$ be these "novel" items with $t = |T|$. As long as the probability that any of these items are output by the algorithm is at most $\frac{\delta}{2}$, $(\varepsilon, \delta)$-DP will be maintained. Consider a single item $i \in T$ which has zero probability of being output by a weight and threshold algorithm run on $\mathcal{S}$ but is given some weight $w'(i)$ when ALG is run on $\mathcal{S}'$. The item will be output only if after adding the Gaussian noise with standard deviation $\sigma$, the noised weight exceeds $\rho$. The probability that any item in $T$ is output follows from a union bound. In order to union bound only over finitely many events, we rely on the fact that $t \leq \Delta_0$; this is why the cardinalities must be capped. This motivates the second important sensitivity measure of ALG.

**Definition 3.2.** The *novel* $\ell_\infty$ *sensitivity* of a weighting algorithm is parameterized by the number $t = |T|$ of items

which are unique to the new user, and is defined as the smallest value $\Delta_\infty(t)$ such that for all possible inputs $\{S_u\}_{u=1}^n$ and new user sets $S_v$,

$$\Delta_\infty(t) \geq \max_{i \in T} w'(i).$$

Then, the calculation of $\rho$ to obtain $(\varepsilon, \delta)$-DP is obtained based on the novel $\ell_\infty$ sensitivity, $\delta$, $\sigma$, and $\Delta_0$. This is formalized in the following theorem whose proof is given in Appendix B.

**Theorem 3.3.** *Let* $\mathcal{S}, (\varepsilon, \delta), \Delta_0,$ ALG, $h$ *be inputs to Algorithm 1. If* ALG *has bounded* $\ell_2$ *and novel* $\ell_\infty$ *sensitivities*

$$\Delta_2 \leq 1 \quad and \quad \Delta_\infty(t) \leq h(t),$$

*then releasing* $U, \tilde{w}_{ext}$ *satisfies* $(\varepsilon, \delta)$-*DP.*

# 4. Maximum Adaptive Degree Weighting

Our main result is an *adaptive* weighting algorithm MaxAdaptiveDegree (MAD) which is amenable to parallel implementations and has the exact same $\ell_2$ and novel $\ell_\infty$ sensitivities as Basic. Therefore, within the weight and threshold meta-algorithm, both algorithms utilize the same noise $\sigma$ and threshold $\rho$ to maintain privacy. Our algorithm improves upon Basic by reallocating weight from items far above the threshold to other items.

We present the full algorithm in Algorithm 2. For simplicity, we will first describe the "unbiased" version of our algorithm where $b, b_{min}, b_{max}$ are set to ones and UserWeights$(S_u, b, b_{min}, b_{max})$ is a vector of weights over all items with $1/\sqrt{|S_u|}$ for every $i \in S_u$ and zeros in other coordinates. The algorithm takes two additional parameters: a maximum adaptive degree $d_{max} \in (1, \Delta_0]$ and an adaptive threshold $\tau = \rho + \beta\sigma$ for a free parameter $\beta \geq 0$. Users with set cardinalities greater than $d_{max}$ are set aside and contribute basic uniform weights to their items at the end of the algorithm. The rest of the users participate in adaptive reweighting. We start from a uniform weighting where each user sends $1/|S_u|$ weight to each of their items. Items have their weights truncated to $\tau$ and any excess weight is sent back to the users proportional to the amount they contributed. Users then reroute a carefully chosen fraction (depending on $d_{max}$) of this excess weight across their items. Finally, each user adds $1/\sqrt{|S_u|} - 1/|S_u|$ to the weight of each of their items.

Each of these stages requires linear work in the size of the input, i.e. the sum of the sizes of the users sets. Furthermore, each stage is straightforward to implement within a parallel framework. As there are a constant number of stages, the algorithm can be implemented with total linear work and constant number of rounds.

---

**Algorithm 2** $\mathrm{MAD}(\mathcal{S}, \tau, d_{max}, b, b_{min}, b_{max})$
**Input:** User sets $\mathcal{S} = \{(u, S_u)\}_{u \in [n]}$, adaptive threshold $\tau \geq 0$, maximum adaptive degree $d_{max} > 1$, biases $b : \mathcal{U} \rightarrow \mathbb{R}$, minimum bias $b_{min} \in [0.5, 1]$, maximum bias $b_{max} \in [1, \infty]$
**Output:** $w : \mathcal{U} \rightarrow \mathbb{R}$ weighting of the items

---

1: Initialize weight vectors $w, w_{init}, w_{trunc}, w_{reroute}$ with zeros.
2: Set reroute discount factor $\alpha = b_{min} - \frac{1}{2\sqrt{d_{max}}}$.
3: $I_{adapt} = \{u \in [n] : \left\lceil \frac{1}{(b_{min})^2} \right\rceil \leq |S_u| \leq d_{max}\}$ ▷ Only users with certain degrees act adaptively.
4: **for all** $u \in I_{adapt}$ **do**
5:     $w_{init}(i) \mathrel{+}= 1/|S_u| \quad \forall i \in S_u$ ▷ Initial $\ell_1$ sensitivity bounded weights.
6: **end for**
7: $r(i) \leftarrow \min\left\{0, \frac{w_{init}(i)-\tau}{w_{init}(i)}\right\}$ for $i \in \mathcal{U}$ ▷ Fraction of weight that exceeds the threshold.
8:
9: $w_{trunc}(i) \leftarrow \min\{w_{init}(i), \tau\}$ for $i \in \mathcal{U}$ ▷ Truncate weights above threshold.
10: **for all** $u \in I_{adapt}$ **do**
11:     $e_u \leftarrow (1/|S_u|) \sum_{i \in S_u} r(i)$ ▷ Excess weight returns to each user proportional to their contribution.
12:     $w_{reroute}(i) \mathrel{+}= \alpha e_u/d_{max}$ for $i \in S_u$ ▷ Reroute excess to items, discounted by $\alpha/d_{max}$.
13: **end for**
14: $w \leftarrow w_{trunc} + w_{reroute}$ ▷ Total $\ell_1$ bounded adaptive weights.
15: $w_b^u \leftarrow \mathrm{UserWeights}(S_u, b, b_{min}, b_{max}) \quad \forall u \in [n]$ ▷ See Algorithm 4.
16: **for all** $u \in I_{adapt}$ **do**
17:     $w(i) \mathrel{+}= w_b^u(i) - 1/|S_u|$ for $i \in S_u$ ▷ Add $\ell_2$ bounded weight and subtract initial weights.
18: **end for**
19: **for all** $u \in [n] \setminus I_{adapt}$ **do**
20:     $w(i) \mathrel{+}= w_b^u(i)$ for $i \in S_u$ ▷ Add $\ell_2$ bounded weight for non-adaptive items.
21: **end for**
22: **return** $w$

---

### 4.1. MAD2R: Biased Weights in Multiple Rounds

This unbiased version of MAD directly improves on the basic algorithm. We further optimize our algorithm by refining an idea from the prior work of DP-SIPS (Swanberg et al., 2023). In that work, the privacy budget is split across multiple rounds with Basic used in each round. In each round, items found in previous rounds are removed from the users' sets, so that in early rounds, easy-to-output (loosely speaking, high frequency) items are output, with more weight being allocated to harder-to-output items in future rounds. The privacy of this approach follows from post-processing: we can freely use the differentially private output $U$ from early rounds to remove items in later rounds.

We propose MaxAdaptiveDegreeTwoRounds (MAD2R) which as a starting point runs MAD in two rounds, splitting the privacy budget as in DP-SIPS. As MAD stochastically dominates Basic, this provides a drop-in improvement. Our key insight comes from the modified meta-algorithm we present in Section 3 which also outputs the vector of noisy weights $\tilde{w}_{ext}$. As long as the ALG maintains bounded sensitivity, we are free to query the noisy weights from prior rounds when constructing weights in future rounds.

We leverage this by running in two rounds with the full pseudocode given in Algorithm 5. In the first round, we run the unbiased version of MAD described above to produce

outputs $U_1 = U$ as well as query access to $\tilde{w}_{ext}$. We will only ever query items in $\mathcal{U}$, so we maintain $\tilde{w}_1$ which is $\tilde{w}_{ext}$ restricted to $\mathcal{U}$ without ever materializing $\tilde{w}_{ext}$. Importantly though, we never release $\tilde{w}_1$ as a final output.

In the second round, we make three preprocessing steps before running MAD. Let $\sigma_1$ be the standard deviation of the noise in the first round and $\rho_2$ be the threshold in the second round. For parameters $C_{lb}, C_{ub} \geq 0$, Let $\tilde{w}_{lb} = \tilde{w}_1 - C_{lb} \cdot \sigma_1$ and $\tilde{w}_{ub} = \tilde{w}_1 + C_{ub} \cdot \sigma_1$ be lower and upper confidence bounds on the true item weights $w_1$ in the first round, respectively.

(a) (DP-SIPS) We remove items from users' sets which belong to $U_1$.

(b) (Ours) We remove items $i$ from users' sets which have weight significantly below the threshold where $\tilde{w}_{ub}(i) < \rho_2$. If $\tilde{w}_{ub}(i)$ is very small, we have little chance of outputting the item in the second round and would rather not waste weight on those items. This is particularly relevant for long-tailed distributions we often see in practice where there are many elements which appear in only one or a few users' sets.

(c) (Ours) For items with $\tilde{w}_{lb} \geq \rho_2$, we assign these items biases $b(i) = \rho_2/\tilde{w}_{lb}(i)$. Via UserWeights (Algorithm 4), we (loosely) try to have each user contribute a $b(i)$ fraction of their normal $1/\sqrt{|S_u|}$ weight while

increasing the weights on unbiased items. As the lower bound on these item weights is very large, we do not need to spend as much of our $\ell_2$ budget on these items. For technical reasons, in order to preserve the overall sensitivity of MAD, we must enforce minimum and maximum bias parameters $b_{min} \in [0.5, 1]$ and $b_{max} \in [1, \infty)$. The weights returned by UserWeights are in the interval $[b_{min}/\sqrt{|S_u|}, b_{max}/\sqrt{|S_u|}]$ and have an $\ell_2$ norm of 1.

## 4.2. Privacy

We state the key lemmas that MAD, when used in the WeightAndThreshold meta-algorithm, is $(\varepsilon, \delta)$-DP. The privacy of MAD2R then follows from basic composition, importantly using our generalized analysis in Section 3 which allows us to compute biased weights from the noisy weights of the first round. The key technical challenge is to bound the $\ell_2$ and novel $\ell_\infty$ sensitivities of MAD. Given space constraints, we defer the proofs to Appendix B.

**Lemma 4.1** (Novel $\ell_\infty$ sensitivity). *Algorithm 2 has novel $\ell_\infty$ sensitivity bounded by $\Delta_\infty(t) \leq \frac{b_{max}}{\sqrt{t}}$.*

**Lemma 4.2** ($\ell_2$ sensitivity). *Algorithm 2 has $\ell_2$-sensitivity upper bounded by 1.*

## 4.3. Utility

In Theorem 1.2, we show that for any input, MAD's performance stochastically dominates Basic: the probability that an item is output by Basic is upper bounded by the probability it is output by MAD. We defer a formal statement of Theorem 1.2 and its proof to Appendix C.

This result captures the worst-case behavior of MAD; it is always at least as good as Basic. MAD can actually do much better as it increases the weight on items below the threshold $\tau$. In Appendix C, we also show a instance where MAD significantly improves upon Basic and DP-SIPS.

# 5. Experiments

| Dataset | Users | Items | Entries |
|---|---|---|---|
| Higgs | $2.8 \times 10^5$ | $5.9 \times 10^4$ | $4.6 \times 10^5$ |
| IMDb | $5.0 \times 10^4$ | $2.0 \times 10^5$ | $7.6 \times 10^6$ |
| Reddit | $2.2 \times 10^5$ | $1.5 \times 10^5$ | $7.9 \times 10^6$ |
| Finance | $1.4 \times 10^6$ | $2.7 \times 10^5$ | $1.7 \times 10^7$ |
| Wiki | $2.5 \times 10^5$ | $6.3 \times 10^5$ | $1.8 \times 10^7$ |
| Twitter | $7.0 \times 10^5$ | $1.3 \times 10^6$ | $2.7 \times 10^7$ |
| Amazon | $4.0 \times 10^6$ | $2.5 \times 10^6$ | $2.4 \times 10^8$ |
| Clueweb | $9.6 \times 10^8$ | $9.4 \times 10^8$ | $4.3 \times 10^{10}$ |
| Common Crawl | $2.9 \times 10^9$ | $1.8 \times 10^9$ | $7.8 \times 10^{11}$ |

Table 1: Number of distinct users, distinct items, and total entries (user, item pairs). The number of entries is the sum of the sizes of all the users' sets.

We now compare the empirical performance of MAD and MAD2R against two parallel (Basic, DP-SIPS) and two sequential algorithms (PolicyGaussian and GreedyUpdate) for the partition selection. We observe that our algorithms output most items (at parity of privacy parameter) among the parallel algorithms for every dataset and across various parameter regimes. Moreover, parallelization allows us to analyze datasets with up to 800 billion entries, orders of magnitude larger than sequential algorithms. In the rest of the section, we describe the datasets, algorithms, and computational setting, before presenting our empirical results.

## 5.1. Datasets

We consider 9 datasets with statistics detailed in Table 1. First, we consider small-scale datasets that are suitable for fast processing by sequential algorithms in a single-core architecture. These includes, for the sake of replicability, datasets used in prior works (Gopi et al., 2020; Carvalho et al., 2022; Swanberg et al., 2023). These datasets have up to 3 million distinct items and 300 million entries. Higgs (Leskovec & Krevl, 2014) is a dataset of Tweets during the discovery of the Higgs. IMDb (Maas et al., 2011) is a dataset of movie reviews, Reddit (Gopi et al., 2020) is a dataset of posts to r/askreddit, Finance (Aenlle) is dataset of financial headlines, Wiki (Wijkhuizen) is a dataset of Wikipedia abstracts, Twitter (Axelbrooke, 2017) is a dataset of customer support tweets, and Amazon (McAuley & Leskovec, 2013; Zhang et al., 2015) is a dataset of product reviews. For each of these text-based datasets we replicate prior methodology (Gopi et al., 2020; Carvalho et al., 2022) where items represent the tokens used in a document and each document corresponds to a user (in some datasets, actual users are tracked across documents, in which case, we use combine the users' documents into one document).

We also consider two very-large publicly-available datasets Clueweb (Boldi et al., 2011) and Common Crawl[1]. The latter has approximately 2 billion distinct items and 800 billion entries. This is 3 orders of magnitude larger than the largest dataset used in prior work. Clueweb (Boldi et al., 2011) is a dataset of web pages and their hyper-links, items corresponds to the hyperlinks on a web page and each page corresponds to a user. Common Crawl is a very-large text dataset of crawled web pages often used in LLM research.

## 5.2. Algorithms and Parameters

We compare our results to both sequential and parallel algorithms from prior work. The sequential algorithms we compare against are PolicyGaussian (Gopi et al., 2020) and GreedyUpdate (Carvalho et al., 2022). Like our algorithm, both algorithms set an adaptive threshold $\tau$ greater than the

---

[1] https://www.commoncrawl.org/

| Dataset | Parallel Algorithms | | | | Sequential Algorithms | |
|---|---|---|---|---|---|---|
| | MAD (ours) | MAD2R (ours) | Basic | DP-SIPS | PolicyGaussian | GreedyUpdate |
| Higgs | **1,807** (±13) | 1,767 (±15) | 1,791 (±18) | 1,743 (±8) | 1,923 (±18) | 2,809 (±11) |
| IMDb | 2,516 (±12) | **3,369** (±19) | 2,504 (±7) | 3,076 (±16) | 3,578 (±19) | 1,363 (±11) |
| Reddit | 4,162 (±19) | **6,215** (±18) | 4,062 (±21) | 5,784 (±30) | 7,170 (±39) | 6,340 (±16) |
| Finance | 12,759 (±16) | **17,785** (±28) | 12,412 (±50) | 16,926 (±18) | 20,100 (±49) | 23,556 (±27) |
| Wiki | 7,812 (±12) | **10,554** (±41) | 7,753 (±36) | 9,795 (±21) | 11,455 (±21) | 4,739 (±14) |
| Twitter | 9,074 (±23) | **14,064** (±13) | 8,859 (±22) | 13,499 (±50) | 15,907 (±30) | 15,985 (±29) |
| Amazon | 35,797 (±63) | **67,086** (±59) | 35,315 (±69) | 66,126 (±57) | 77,846 (±127) | 86,841 (±95) |
| Clueweb | 34,692,178 | 34,533,524 | 34,603,077 | **34,889,208** | – | – |
| Common Crawl | 15,815,452 | **29,373,829** | 15,734,148 | 28,328,613 | – | – |

Table 2: Comparison of output size of DP partition selection algorithms with $\varepsilon = 1$, $\delta = 10^{-5}$, and $\Delta_0 = 100$. A standard hyperparameter setting is fixed for each algorithm, other than DP-SIPS, where the best result is taken from privacy splits $[0.1, 0.9]$ and $[0.05, 0.15, 0.8]$. For smaller datasets, sequential algorithms are also reported as oracles and results are averaged over 5 trials with one standard deviation reported parenthetically. For each dataset, the best parallel result is bolded and the best sequential result is underlined.

true threshold $\rho$. They try to maximize weight assigned to items up to but not exceeding $\tau$. PolicyGaussian goes through each user set one by one and adds $\ell_2$ bounded weight to minimize the $\ell_2$ distance between the current weight and the all $\tau$ vector, $w(i) = \tau \ \forall i \in \mathcal{U}$. GreedyUpdate goes through each user set one by one and increments the weight of a single item in the set by one, choosing an item whose weight is currently below $\tau$.[2] As observed before (Swanberg et al., 2023), sequential algorithms can have arbitrary long adpativity chains (the processing of each user can depend on all prior users processed) thus allowing larger output sizes than parallel algorithms. This, however, comes at the cost of not being parallelizable (as we observe in our experiments on the larger datasets). The parallel baselines we compare against are Basic (Korolova et al., 2009; Gopi et al., 2020) and DP-SIPS (Swanberg et al., 2023). In DP-SIPS, the privacy budget is split into a distribution over rounds. In each round, the basic algorithm is run with the corresponding privacy budget. Items found in previous rounds are removed from all user's sets for the next rounds.

We make parameter choices which are consistent with prior work and generally work well across datasets (see Appendix D for more parameter settings). Unless otherwise specified, we use $\varepsilon = 1$, $\delta = 10^{-5}$, and $\Delta_0 = 100$.[3] For PolicyGaussian and GreedyUpdate, we set the $\beta = 4$ to be the number of standard deviations of noise to add to the base threshold to set the adaptive threshold. For DP-SIPS, we take the best result of running with a privacy split of $[0.1, 0.9]$ and $[0.05, 0.15, 0.8]$.[4] For MAD and MAD2R, we

set $d_{max} = 50$ and $\beta = 2$. For MAD2R, we set the privacy split of $[0.1, 0.9]$, $b_{min} = 0.5$, $b_{max} = 2$, $C_{lb} = 1$, and $C_{ub} = 3$.

### 5.3. Computing Details

We perform experiments in two different computational settings. First we implement a sequential, in-memory version of all algorithms (including the parallel ones) using Python. For PolicyGaussian and GreedyUpdate we use the Python implementations from prior work (Gopi et al., 2020; Carvalho et al., 2022). This allows us to fairly test the scalability of the algorithms not using parallelism. As we observe next, this approach does not scale to the two largest datasets we have (Clueweb, Common Crawl).

Then, we implement all parallel algorithms (MAD, MAD2R, Basic, DP-SIPS) using C++ in a modern multi-machine massively parallel computation framework in our institution. This framework allows to use a fleet of shared (x86_64) architecture machines with 2.45GHz clocks. The machines are shared by several projects and can have up to 256 cores and up to 512GB of RAM. The jobs are dynamically allocated RAM, machines and cores depending on need and availability. As we observe, all parallel algorithm are very scalable and run on these huge datasets within 4 hours of wall-clock time. On the other hand, both sequential algorithms cannot exploit this architecture and could not complete in 16 hours on the Clueweb dataset (we estimate they would take several days to complete on the Common Crawl dataset even assuming access to enough memory).

### 5.4. Results

**Algorithm Comparison**  Table 2 displays the output size of the DP partition selection algorithms (i.e., the number of privatized items output). Among parallel algorithms, MAD2R achieves the best result on seven out of nine datasets. The two exceptions are the Higgs dataset, where MAD performs the best, and the Clueweb dataset, where DP-SIPS

---

[2]Unlike all of the other algorithm, this algorithm does not do a first step of bounding users' degrees by $\Delta_0$ as it only assigns weight to a single item per user by design.

[3]We report these privacy settings for consistency with prior work in the literature, but observe the results are consistent across various choices. For real production deployments on large-scale sensitive data, $\delta$ is usually smaller.

[4]As this choice can have a significant effect on performance, we choose the best-performing to give this baseline the benefit of the doubt.

performs the best. Both of these datasets have outlier statistics (see Table 1): the average size of a user set in the Higgs dataset is less than 2 and the number of unique items in the Clueweb dataset is less than the number of users. Directly comparing MAD with Basic, MAD is always better, corroborating our proof of stochastic dominance. Comparing MAD2R with DP-SIPS, MAD2R is almost always significantly better, by up to a factor of a $9.5\%$ improvement on the IMDb dataset.

On the small scale datasets where we can run sequential algorithms, as expected from prior work (Swanberg et al., 2023), one of the two sequential algorithms yield the best results across all algorithms with PolicyGaussian consistently outperforming all parallel baselines. GreedyUpdate's performance is heavily dataset dependent, sometimes performing the best and sometimes the worst out of all algorithms. This is not a surprise as the sequential algorithms utilize much more adaptivity than even our adaptive parallel algorithm at the cost of limiting scalability. Our algorithm is still competitive, never outputting fewer than $86\%$ of the items of PolicyGaussian (and outperforming GreedyUpdate on many datasets). For massive datasets, where it is simply infeasible to run the sequential algorithms, however MAD2R has the best results of all parallel algorithms.

Figures comparing output sizes while varying $\varepsilon$, $\delta$, $\Delta_0$, and $d_{max}$ are included in Appendix D. The relative performance of the algorithms is the same across many choices.

**Absolute Utility** To understand the absolute utility of our algorithms (as opposed to relative to other baselines), we focus on the performance of MAD2R on the Reddit and Common Crawl datasets. In order to understand the performance in a real deployment rather than compare baselines across common parameter settings, we change the $\delta$ for the large scale Common Crawl dataset to $\delta = 10^{-11}$.

On the Reddit dataset, MAD2R outputs 6,340 out of 143,556 unique items ($4.4\%$). On the other hand, $98\%$ of users have at least one outputted item, and $45\%$ of the entries (user-item pairs) belong to an item which is output by our algorithm. The relatively small overall fraction of items output is due in part to the fact that the Reddit dataset has $58\%$ singleton items (items only appearing in a single user's set). Any algorithm which outputs any singleton items is not private, as it is leaking private information belonging to a single user. For any algorithm with acceptable privacy settings, outputting items with very small frequencies is also simply not possible. In Figure 1a, we break down the number of items total in the dataset and the number output by MAD2R broken down by item frequency. Our algorithm returns almost all of the items with frequency at least 100.

On the Common Crawl dataset, MAD2R outputs 16,551,550 out of 1,803,720,630 unique items ($0.9\%$). On the other

hand, $99.9\%$ of users have at least one outputted item and $97\%$ of entries in the dataset belong to an item in output by our algorithm. This dataset contains $61\%$ singleton items, and many low frequency items. In Figure 1b, we break down the number of items total in the dataset and the number output by MAD2R broken down by item frequency. Our algorithm returns an overwhelming fraction of items occurring in at least 200 user sets on a dataset with billions of users overall.

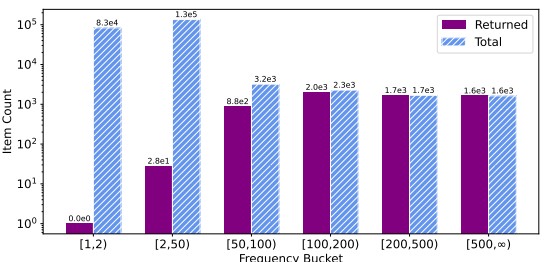

(a) Reddit Item Coverage

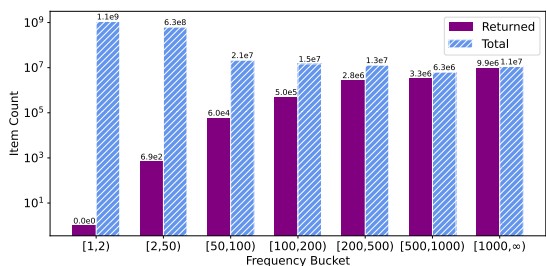

(b) Common Crawl Item Coverage

Figure 1: Comparison by item frequency of the output size of MAD2R to the total items on the Reddit and Common Crawl datasets. Parameters $\varepsilon = 1$ and $\Delta_0 = 100$ are fixed with $\delta = 10^{-5}$ for Reddit and $\delta = 10^{-11}$ for Common Crawl.

## 6. Conclusion

We introduce MAD and MAD2R, new parallel algorithms for private partition selection which provide state-of-the-art results, scale to massive datasets, and provably outperform baseline algorithms. Closing the remaining gap between parallel and sequential algorithms remains an interesting direction, as well as developing new ideas to adaptive route weight to items below the privacy threshold while maintaining bounded sensitivity. While we are able to prove *ordinal* theoretical results (our algorithm is at least as good as another), it is an open challenge to develop a framework where we can prove *quantitative* results, perhaps comparing the competitive ratio of a private partition selection algorithm compared to some reasonably defined optimum.

## Acknowledgements

Justin Chen is supported by an NSF Graduate Research Fellowship under Grant No. 17453. Part of this work was done while he was a student researcher at Google Research.

## Impact Statement

This paper presents work whose goal is to advance the field of Machine Learning. There are many potential societal consequences of our work, none which we feel must be specifically highlighted here.

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

# A. Algorithm Pseudocode

## A.1. Basic Pseudocode

---

**Algorithm 3** `Basic`
**Input:** User sets $\mathcal{S} = \{(u, S_u)\}_{u \in [n]}$
**Output:** $w : \mathcal{U} \to \mathbb{R}$ weighting of the items

---

1: Initialize weight vector $w$ with zeros
2: **for all** $u \in [n]$ **do**
3:     $w(i) \mathrel{+}= 1/\sqrt{|S_u|}$ for $i \in S_u$                                             ▷ Add basic $\ell_2$ bounded weight.
4: **end for**
5: **return** $w$

---

## A.2. MAD and MAD2R Pseudocode

See Algorithm 2 in the main text for the pseudocode of `MAD`. Here, we include the pseudocode for the subroutine `UserWeights` and our two-round algorithm `MAD2R`.

---

**Algorithm 4** `UserWeights`$(S_u, b, b_{min}, b_{max})$
**Input:** User set $S_u \subseteq \mathcal{U}$, biases $b : \mathcal{U} \to [0, 1]$, minimum bias $b_{min} \in [0.5, 1]$, maximum bias $b_{max} \in [1, \infty]$
**Output:** $w_b : S_u \to \mathbb{R}$ weighting of the items

---

1: Initialize weight vector $w_b$ with zeros
2: $S_{biased} = \{i \in S_u : b(i) < 1\}$
3: $S_{unbiased} = S_u \setminus S_{biased}$
4: $w_b(i) \leftarrow \frac{\max\{b_{min}, b(i)\}}{\sqrt{|S_u|}}$ for $i \in S_{biased}$                  ▷ Set biased weights, respecting min bias
5: $w_b(i) \leftarrow \min\left\{ \frac{b_{max}}{\sqrt{|S_u|}}, \sqrt{\frac{1 - \sum_{i \in S_{biased}} w_b(i)^2}{|S_{unbiased}|}} \right\}$ for $i \in S_{unbiased}$     ▷ Allocate remaining $\ell_2$ budget, respecting max bias
6: **while** $\sum_{i \in S_u} w_b(i)^2 < 1$ **do**
7:     $S_{small} \leftarrow \left\{ i \in S_u : w_b(i) < \frac{1}{\sqrt{|S_u|}} \right\}$
8:     $C \leftarrow \min\left\{ \frac{b_{max}/\sqrt{|S_u|}}{\max_{i \in S_{small}} w_b(i)}, \sqrt{1 + \frac{1 - \sum_{i \in S_u} w_b(i)^2}{\sum_{i \in S_{small}} w_b(i)^2}} \right\}$
9:     $w_b(i) \leftarrow C \cdot w_b(i)$ for $i \in S_{small}$         ▷ Increase small weights using remaining $\ell_2$ budget, respecting max bias
10: **end while**
11: **return** $w_b$

---

---

**Algorithm 5** $\texttt{MAD2R}(\mathcal{S}, (\varepsilon_1, \delta_1), (\varepsilon_2, \delta_2), \Delta_0, d_{max}, \beta, C_{lb}, C_{ub}, b_{min}, b_{max})$

**Input:** User sets $\mathcal{S} = \{(u, S_u)\}_{u \in [n]}$, privacy parameters $(\varepsilon_1, \delta_1)$ and $(\varepsilon_2, \delta_2)$, degree cap $\Delta_0$, maximum adaptive degree $d_{max}$, adaptive threshold excess parameter $\beta$, lower bound constant $C_{lb}$, upper bound constant $C_{ub}$, minimum bias $b_{min}$, maximum bias $b_{max}$

**Output:** Subset of the union of user sets $\mathcal{U} = \cup_{u=1}^{n} S_u$

---

1: For $u \in [n]$, cap $S_u$ to at most $\Delta_0$ items by random subsampling
2: Select $\sigma_r$ corresponding to the Gaussian Mechanism (Proposition 2.4) for $(\varepsilon_r, \delta_r/2)$-DP with $\Delta_2 = 1$ for $r \in \{1, 2\}$.
3: **Round 1**
4: Set threshold $\rho_1 = \max_{t \in [\Delta_0]} \frac{1}{\sqrt{t}} + \sigma_1 \Phi^{-1}\left( \left(1 - \frac{\delta}{2}\right)^{1/t} \right)$
5: $w_1 \leftarrow \texttt{MAD}(\mathcal{S}, \rho_1 + \beta\sigma_1, d_{max}, \overrightarrow{1}, 1, 1)$      $\triangleright$ Compute $\texttt{MAD}$ (Algorithm 2) weights in the first round.
6: $\tilde{w}_1 \leftarrow w_1 + \mathcal{N}(0, \sigma_1^2 I)$      $\triangleright$ Add noise
7: $U_1 \leftarrow \{i \in \mathcal{U} : \tilde{w}_1(i) \geq \rho_1\}$      $\triangleright$ Apply threshold
8: **Round 2**
9: Set threshold $\rho_2 = \max_{t \in [\Delta_0]} \frac{b_{max}}{\sqrt{t}} + \sigma_2 \Phi^{-1}\left( \left(1 - \frac{\delta}{2}\right)^{1/t} \right)$
10: $\tilde{w}_{lb} \leftarrow \max\{0, \tilde{w}_1 - C_{lb} \cdot \sigma_1\}$      $\triangleright$ Weight lower bound from Round 1
11: $\tilde{w}_{ub} \leftarrow \tilde{w}_1 + C_{ub} \cdot \sigma_1$      $\triangleright$ Weight upper bound from Round 1
12: $U_{low} \leftarrow \{i \in \mathcal{U} : \tilde{w}_{ub} < \rho_2\}$
13: $S_u \leftarrow S_u \setminus (U_1 \cup U_{low})$ for $u \in [n]$      $\triangleright$ Remove items found in Round 1 or with a small upper bound on the weight
14: $b \leftarrow \min\left\{1, \frac{\rho_2}{\tilde{w}_{lb}}\right\}$      $\triangleright$ Bias weights to not overshoot threshold
15: $w_2 \leftarrow \texttt{MAD}(\mathcal{S}, \rho_2 + \beta\sigma_2, d_{max}, b, b_{min}, b_{max})$      $\triangleright$ Compute $\texttt{MAD}$ (Algorithm 2) in the second round
16: $\tilde{w}_2 \leftarrow w_2 + \mathcal{N}(0, \sigma_2^2 I)$      $\triangleright$ Add noise
17: $U_2 \leftarrow \{i \in \mathcal{U} : \tilde{w}_2(i) \geq \rho_2\}$      $\triangleright$ Apply threshold
18: **return** $U_1 \cup U_2$

---

# B. Proof of Privacy

## B.1. Meta-algorithm

We start by proving the privacy of the meta-algorithm described in Section 3.

*Proof of Theorem 3.3.* Let $w_{ext} : \Sigma \to \mathbb{R}$ be an extension of the weight vector $w$ returned by ALG where

$$w_{ext}(i) = \begin{cases} w(i) & \text{if } i \in \mathcal{U} \\ 0 & \text{if } i \in \Sigma \setminus \mathcal{U} \end{cases}.$$

Note that $\tilde{w}_{ext}$ is exactly the result of applying the Gaussian Mechanism to $w_{ext}$. Furthermore, the $\ell_2$ sensitivity (Definition 3.1) of computing $w_{ext}$ is the same as that of $w$ as items outside of $\mathcal{U}'$ do not contribute to the sensitivity as they are 0 regardless of whether the input is $\mathcal{S}$ or $\mathcal{S}'$. By the choice of $\sigma$ according to Proposition 2.4 with $\Delta_2 \leq 1$, releasing $\tilde{w}_{ext}$ is $\left(\varepsilon, \frac{\delta}{2}\right)$-DP.

The privacy of releasing $U$ depends on our choice of the threshold $\rho$. We will first show that the probability that any of $t$ i.i.d. draws from a Gaussian random variable $\mathcal{N}(0, \sigma^2)$ exceeds $\sigma\Phi^{-1}\left(\left(1 - \frac{\delta}{2}\right)^{1/t}\right)$ is exactly $\frac{\delta}{2}$. Let $A$ be the bad event, $Y \sim \mathcal{N}(0, \sigma^2)$, and $Z \sim \mathcal{N}(0, 1)$:

$$\begin{aligned}
\Pr(A) &= 1 - \Pr\left(Y \leq \sigma\Phi^{-1}\left(\left(1 - \frac{\delta}{2}\right)^{1/t}\right)\right)^t \\
&= 1 - \Pr\left(Z \leq \Phi^{-1}\left(\left(1 - \frac{\delta}{2}\right)^{1/t}\right)\right)^t \\
&= 1 - \Phi\left(\Phi^{-1}\left(\left(1 - \frac{\delta}{2}\right)^{1/t}\right)\right)^t \\
&= 1 - \left(1 - \frac{\delta}{2}\right)^{t/t} \\
&= \frac{\delta}{2}.
\end{aligned}$$

By the condition that $h(t)$ is an upper bound on $\Delta(t)$, the choice of $\rho$ implies that, no matter how many items are novel (unique to the new user in a neighboring dataset), the probability that any of them belong to $U$ is at most $\frac{\delta}{2}$. Conditioned on the release of $\tilde{w}_{ext}$, releasing $U$ is $\left(0, \frac{\delta}{2}\right)$-DP. By basic composition, the overall release is $(\varepsilon, \delta)$-DP, as required. □

## B.2. MADW and MADW2R

We now prove the privacy of our main algorithm by bounding its $\ell_2$ and novel $\ell_\infty$ sensitivities.

**Theorem B.1** (Privacy of MAD2R). *Algorithm 5 is $(\varepsilon_1 + \varepsilon_2, \delta_1 + \delta_2)$-DP.*

The rest of this section will be devoted to proving this theorem. We first state as a corollary that MAD run in a single round without biases is also private.

**Corollary B.2** (Privacy of MAD). *Releasing the output $U$ from Algorithm 1 run with unbiased MAD (Algorithm 2) as the weighting algorithm and with $h(t) = \frac{1}{\sqrt{t}}$ is $(\varepsilon, \delta)$-DP.*

*Proof.* This follows directly from Theorem B.1 by setting $(\epsilon_1, \delta_1) = (\epsilon, \delta)$ and $(\epsilon_2, \delta_2) = (0, 0)$. □

To prove privacy, consider a weight vector $w$ returned by Algorithm 2 for an input $\mathcal{S} = \{(u, S_u)\}_{u=1}^n, \tau, d_{max}, b, b_{min}, b_{max}$ and the output $w'$ for an input $\mathcal{S}' = \mathcal{S} \cup \{(v, S_v)\}, \tau, d_{max}, b', b_{min}, b_{max}$ which includes a new user $v$ not in the original input. Let $d_v$ be the degree of the new user. Let $T = S_v \setminus \cup_{u=1}^n S_u$ be the subset of items which appear only in $S_v$ and not in any of the original user sets, and let $t = |T|$. The vectors of biases $b$ and $b'$ are defined on the set of items $\cup_{u=1}^n S_u$ and

$(\cup_{u=1}^{n} S_u) \cup T$ respectively. We remind the notation used in Algorithm 2 for vector $w_b^v$ to denote the biased weight vector the new user $v$ computes by calling Algorithm 4, i.e. `UserWeights`.

**Lemma B.3** (Novel $\ell_\infty$ sensitivity with biases). *Assume that the adaptive threshold is $\tau \geq 1$, and the maximum adaptive degree is $d_{max} \geq 4$. Then, Algorithm 2 has novel $\ell_\infty$ sensitivity bounded by*

$$\Delta_\infty(t) \leq \frac{b_{max}}{\sqrt{t}}.$$

*where $t = |T|$ is the cardinality of $T$, the set of novel items.*

*Proof.* Unpacking Definition 3.2, it suffices to show that

$$\max_{i \in T} w'(i) \leq \frac{b_{max}}{\sqrt{t}}.$$

Note that this is trivially true if $d_v = |S_v| > d_{max}$ or $d_v < \left\lceil \frac{1}{(b_{min})^2} \right\rceil$ since $v$ does not participate in adaptivity due to its too low or high degree. We will proceed by assuming this is not the case.

Consider the final weight of an item $i$ in the set of novel items $T \subseteq S_v$:

$$w'(i) = w'_{trunc}(i) + w'_{reroute}(i) + w_b^v(i) - w'_{init}(i).$$

Note that for all novel items $i \in T$, $w'_{init}(i) = \frac{1}{d_v} \leq \tau$ as $v$ is the sole contributor to the weight of item $i$. Therefore, no weight is truncated or rerouted from these items: $w'_{trunc}(i) = w'_{init}(i) = \frac{1}{d_v}$. Expanding the definition of rerouted weight,

$$
\begin{aligned}
w'_{reroute}(i) &= \frac{\alpha e'_v}{d_{max}} \\
&= \frac{\alpha}{d_{max} d_v} \sum_{j \in S_v \setminus T} r'(j) \\
&= \frac{\alpha}{d_{max} d_v} \sum_{j \in S_v \setminus T} \frac{w'_{init}(j) - w'_{trunc}(j)}{w'_{init}(j)} \\
&\leq \frac{\alpha}{d_{max} d_v} (d_v - t) \\
&= \frac{\alpha}{d_{max}} \left( 1 - \frac{t}{d_v} \right).
\end{aligned}
$$

Finally, note that $w_b^v(i) \leq \frac{b_{max}}{\sqrt{d_v}}$ by construction (this is the meaning of $b_{max}$).

We can bound $w'(i)$ as

$$
\begin{aligned}
w'(i) &= w'_{trunc}(i) + w'_{reroute}(i) + w_b^v(i) - w'_{init}(i) \\
&\leq \frac{1}{d_v} + \frac{\alpha}{d_{max}} \left( 1 - \frac{t}{d_v} \right) + \frac{b_{max}}{\sqrt{d_v}} - \frac{1}{d_v} \\
&= \frac{\alpha}{d_{max}} \left( 1 - \frac{t}{d_v} \right) + \frac{b_{max}}{\sqrt{d_v}}. \tag{1}
\end{aligned}
$$

In the rest of the proof, we will show that the upper bound Equation (1) is maximized when $d_v = t$, i.e., when the first term is zero. Recall that $t \leq d_v \leq d_{max}$. Consider the partial derivative with respect to $d_v$:

$$\frac{\partial}{\partial d_v} \left( \frac{\alpha}{d_{max}} \left( 1 - \frac{t}{d_v} \right) + \frac{b_{max}}{\sqrt{d_v}} \right) = \frac{\alpha}{d_{max}} \frac{t}{d_v^2} - \frac{b_{max}}{2 d_v^{3/2}}.$$

Consider the condition of the derivative being non-positive:

$$0 \geq \frac{\alpha}{d_{max}} \frac{t}{d_v^2} - \frac{b_{max}}{2d_v^{3/2}}$$

$$\iff b_{max} \geq \frac{2\alpha t}{d_{max}\sqrt{d_v}}$$

$$\impliedby b_{max} \geq \frac{2\alpha\sqrt{t}}{d_{max}}$$

$$\impliedby b_{max} \geq \frac{2\alpha}{\sqrt{d_{max}}}.$$

The final condition holds as $\alpha \leq 1$ and by the assumption that $d_{max} \geq 4$. We note that $b_{max}$ is always set to be at least 1. As the derivative is non-positive, the right side of Equation (1) is maximized when $d_v$ is minimized at $d_v = t$. Then, $\Delta_\infty(t) \leq \frac{b_{max}}{\sqrt{t}}$, as required. $\qquad\square$

Following we state some properties of the biased weights $w_b$ which will be helpful in the proof.

**Lemma B.4.** *Let $S_u, b, b_{min}, b_{max}$ be valid inputs to Algorithm 4, and let $w_b$ be the weight vector returned by the algorithm. Let $d = |S_u|$. Then, the following hold:*

- $w_b(i) = 0$ *for all $i \in \mathcal{U} \setminus S_u$*

- $\frac{b_{min}}{\sqrt{d}} \leq w_b(i) \leq \frac{b_{max}}{\sqrt{d}}$ *for all $i \in S_u$*

- $\|w_b\|_2 \leq 1$

*Proof.* The first claim holds as the weight vector is initialized with zeros and only indices $i \in S_u$ are updated by the algorithm.

To simplify the notation, we define $d = |S_u| = |S_{biased}| + |S_{unbiased}|$. As $b(i) \leq 1$ for all $i \in \mathcal{U}$, the initial weights given to items in $S_{biased}$ are between $\frac{b_{min}}{\sqrt{d}}$ and $\frac{1}{\sqrt{d}}$. We also know that $\frac{1}{\sqrt{d}} \leq \frac{b_{max}}{\sqrt{d}}$. The sum of squares of these weights will thus be between $\frac{b_{min}^2 |S_{biased}|}{d}$ and $\frac{|S_{biased}|}{d}$. Call this value $k$.

Weights of items in $S_{unbiased}$ are given by

$$\min\left\{ \frac{b_{max}}{\sqrt{d}}, \sqrt{\frac{1-k}{|S_{unbiased}|}} \right\}$$

We show that the minimum of these two terms is at least $\frac{1}{\sqrt{d}} \geq \frac{b_{min}}{\sqrt{d}}$. The first term is at least $\frac{1}{\sqrt{d}}$ since $b_{max} \geq 1$. To observe the same for the second term, one should plugg in the upper bound of $k \leq \frac{|S_{biased}|}{d}$. By construction, the weights are upper bounded by $\frac{b_{max}}{\sqrt{d}}$. Furthermore, note that the sum of squares of the entire weight vector at this point is upper bounded by 1. In particular, it is equal to 1 for the second term of the minimization:

$$k + \sum_{i \in S_{biased}} \left( \frac{1-k}{|S_{unbiased}|} \right) = k + (1-k) = 1.$$

In the remainder of the algorithm, sum of the weights of items in $S_{small} \subseteq S_{unbiased}$ may increase if the $\ell_2$ norm of the weight vector is strictly less than 1. Consider the weights after any such update by a multiplicative factor $C$ defined as

$$C = \min\left\{ \frac{b_{max}/\sqrt{d}}{\max_{i \in S_{small}} w_b(i)}, \sqrt{1 + \frac{1 - \sum_{i \in S_u} w_b(i)^2}{\sum_{i \in S_{small}} w_b(i)^2}} \right\}.$$

Note that $C > 1$ by definition of $S_{small}$ and the stopping criteria of the while loop. Therefore, none of the final weights will be less than $\frac{b_{min}}{\sqrt{d}}$. Consider the first case of the minimization. Any updated weight $C \cdot w_b(i)$ for $i \in S_{small}$ will be at most

$\frac{b_{max}}{\sqrt{d}}$ as

$$\frac{b_{max}/\sqrt{d}}{\max_{j \in S_{small}} w_b(j)} \cdot w_b(i) \le \max_{i^* \in S_{small}} \frac{b_{max}/\sqrt{d}}{w_b(i^*)} w_b(i^*) = b_{max}/\sqrt{d}.$$

Now, consider the second case of the maximization. Then, the squared $\ell_2$ norm of the weight vector will be

$$\sum_{i \in S_u \setminus S_{small}} w_b(i)^2 + \sum_{i \in S_{small}} \left(1 + \frac{1 - \sum_{j \in S_u} w_b(j)^2}{\sum_{j \in S_{small}} w_b(j)^2}\right) w_b(i)^2 = \left(\sum_{i \in S_u} w_b(i)^2\right) + \left(1 - \sum_{j \in S_u} w_b(j)^2\right) = 1.$$

As $C$ is taken to be the minimum of these two values, the final weight vector will satisfy all of the required bounds. □

We will prove a useful fact that Algorithm 2 is monotone in the sense that weights when run on $\mathcal{S}'$ will only increase compared to when run on only $\mathcal{S}$. We apply this proposition in upper bounding the $\ell_2$ sensitivity of Algorithm 2 in Lemma B.6.

**Proposition B.5** (Monotonicity). *For all $i \in \cup_{u \in \{1,...,n,v\}} S_u$,*

$$w'_{init}(i) \ge w_{init}(i)$$
$$w'_{trunc}(i) \ge w_{trunc}(i)$$
$$w'_{reroute}(i) \ge w_{reroute}(i).$$

*Proof.* Fix any item $i$. As all increments to the initial $\ell_1$ bounded weights are positive and non-adaptive,

$$w'_{init}(i) \ge w_{init}(i).$$

In fact, the two weights are either equal or differ by a factor of $1/d_v$ depending on whether $i \in S_v$. The calculations of the fraction of excess weight that exceeds the threshold, the truncated weights, the excess weight returned to each user, and the rerouted weights are all monotonically non-decreasing with the initial weights. Therefore,

$$w'_{trunc}(i) \ge w_{trunc}$$

and

$$w'_{reroute}(i) \ge w_{reroute}.$$

□

**Lemma B.6** ($\ell_2$ sensitivity with biased weights). *Algorithm 2 has $\ell_2$-sensitivity upper bounded by $1$.*

*Proof.* Note that this is trivially true by *Lemma B.4* if $|S_v| = d_v > d_{max}$ or $d_v < \left\lceil \frac{1}{(b_{min})^2} \right\rceil$ since $v$ does not participate in adaptivity in this case. We will proceed by assuming this is not the case.

Our goal is to bound the $\ell_2$ norm of the difference $\Delta = w' - w$, the difference in final weights with and without the new user $v$. We will use the notation $\Delta_{subscript} = w'_{subscript} - w_{subscript}$. Note that $\Delta_{reroute} = w'_{reroute} - w_{reroute}$ is the additional rerouted weight after adding user $v$ and let $\Delta_{user} = \Delta - \Delta_{reroute}$ be the rest of the difference. Note that $\Delta_{user}$ is $d_v$-sparse and only has nonzero entries on $S_v$, the items of the new user. Our goal will be to bound the $\ell_2$ norms of $\Delta_{reroute}$ and $\Delta_{user}$, thus bounding the $\ell_2$ sensitivity of the algorithm by triangle inequality.

We start by tracking the excess weight created by $v$ which will be useful in bounding the $\ell_2$ norms of both $\Delta_{reroute}$ and $\Delta_{user}$. It is the total amount of weight added by $v$ to items that exceed the threshold[5]:

$$\gamma = \|\Delta_{init} - \Delta_{trunc}\|_1 \qquad (2)$$

(Note that this is not the same as $e'_v$ which is the amount of weight from $v$ gets returned to reroute.)

---

[5]If if an item $i$ only exceeds the threshold due to the addition of $v$, we only consider the allocated weight to $i$ above the threshold.

The total amount of weight that is returned to users to reroute is equal to the amount of weight truncated, i.e., the sum of $w_{init} - w_{trunc}$:

$$\sum_{u=1}^{n} e_u = \sum_{u=1}^{n} (1/|S_u|) \sum_{i \in S_u} r(i)$$

$$= \sum_{u=1}^{n} (1/|S_u|) \sum_{i \in S_u} \max \left\{ 0, \frac{w_{init}(i) - \tau}{w_{init}(i)} \right\}$$

$$= \sum_{u=1}^{n} (1/|S_u|) \sum_{i \in S_u} \frac{w_{init}(i) - w_{trunc}(i)}{w_{init}(i)}$$

$$= \sum_{u=1}^{n} (1/|S_u|) \sum_{i \in S_u} \frac{w_{init}(i) - w_{trunc}(i)}{\sum_{w:i \in S_w} 1/|S_w|}$$

$$= \sum_{i \in \mathcal{U}} \frac{(w_{init}(i) - w_{trunc}(i)) \sum_{u:i \in S_u} 1/|S_u|}{\sum_{w:i \in S_w} 1/|S_w|}$$

$$= \sum_{i \in \mathcal{U}} w_{init}(i) - w_{trunc}(i).$$

For notational parsimony, let $e_v = 0$ (as $v$ does not appear in the original input $\mathcal{S}$). Note that $e_u$ is monotonically increasing with $w_{init}$: if any coordinate of the initial weight increases, the excess ratio of any user will never decrease. With monotonicity of $w_{init}$ from Proposition B.5, it follows that $w'_{init} - w'_{trunc} \geq w_{init} - w_{trunc}$ since the threshold $\tau$ in the capping formula $w_{trunc}(i) \leftarrow \min\{w_{init}(i), \tau\}$ stays the same after adding the new user. Consequently, we have:

$$\gamma = \|(w'_{init} - w'_{trunc}) - (w_{init} - w_{trunc})\|_1$$

$$= \left( \sum_{i \in \mathcal{U}'} w'_{init} - w'_{trunc} \right) - \left( \sum_{i \in \mathcal{U}} w_{init} - w'_{trunc} \right)$$

$$= \sum_{u \in \{1, \dots, n, v\}} e'_u - e_u.$$

Now, we will consider $\Delta_{reroute}$. Recall that $|S_u| \leq d_{max}$ for all $u$ participating in adaptivity. For all other users, the terms $e_u$ and $e'_u$ are zero.

$$\|\Delta_{reroute}\|_1 = \sum_{u \in \{1, \dots, n, v\}} \sum_{i \in S_u} (\alpha/d_{max})(e'_u - e_u)$$

$$= \sum_{u \in \{1, \dots, n, v\}} |S_u|(\alpha/d_{max})(e'_u - e_u)$$

$$\leq \sum_{u \in \{1, \dots, n, v\}} \alpha(e'_u - e_u)$$

$$= \alpha\gamma.$$

Furthermore, we can bound the $\ell_\infty$ norm of $\Delta_{reroute}$ as:

$$w'_{reroute}(i) - w_{reroute}(i) \leq (\alpha/d_{max}) \sum_{u \in \{1, \dots, n, v\}} e'_u - e_u = \alpha\gamma/d_{max}.$$

By Hölder's inequality,

$$\|\Delta_{reroute}\|_2 \leq \sqrt{\|\Delta_{reroute}\|_1 \|\Delta_{reroute}\|_\infty} = \sqrt{\alpha^2\gamma^2/d_{max}} = \frac{\alpha\gamma}{\sqrt{d_{max}}}. \tag{3}$$

Consider the rest of the difference $\Delta_{user}$. For $i \in S_v$, a single coordinate of $\Delta_{user}$ will be comprised of the sum

$$\Delta_{user}(i) = \Delta_{trunc}(i) + w_b^v(i) - 1/d_v.$$

Note that $\Delta_{trunc}(i) \in [0, 1/d_v]$, so
$$\Delta_{user}(i) \in [w_b^v(i) - 1/d_v, w_b^v(i)].$$

Furthermore, as $\Delta_{init}(i) = 1/d_v$,

$$\|\Delta_{user}\|_1 = \sum_{i \in S_v} w_b^v(i) + \Delta_{trunc}(i) - \Delta_{init}(i) = \left(\sum_{i \in S_v} w_b^v(i)\right) - \gamma.$$

Let $x : S_v \to \mathbb{R}$ and $y : S_v \to \mathbb{R}$ be two sets of weights over $S_v$ such that $x(i) = w_b^v(i) - 1/d_v$, $y(i) \in [0, 1/d_v]$, and $\sum_{i \in S_v} x(i) + y(i) = \left(\sum_{i \in S_v} w_b^v(i)\right) - \gamma$. Then,

$$\|\Delta_{user}\|_2 \leq \max_y \|x + y\|_2$$

as any valid $\Delta_{user}$ can be expressed as the sum of $x$ and a choice of $y$ satisfying the above constraints. Note that

$$\|y\|_1 = \left(\sum_{i \in S_v} w_b^v(i)\right) - \gamma - \sum_{i \in S_v} x(i) = \left(\sum_{i \in S_v} w_b^v(i)\right) - \gamma - \sum_{i \in S_v} w_b^v(i) + 1 = 1 - \gamma$$

and by Hölder's inequality,

$$\|y\|_2 \leq \sqrt{\|y\|_1 \|y\|_\infty} = \sqrt{\frac{1 - \gamma}{d_v}}.$$

Then,

$$\begin{aligned}
\|x + y\|_2^2 &= \sum_{i \in S_v} \left(w_b^v(i) - 1/d_v + y(i)\right)^2 \\
&= \sum_{i \in S_v} w_b^v(i)^2 + \frac{1}{d_v^2} + y(i)^2 - \frac{2w_b^v(i)}{d_v} + 2y(i) \cdot w_b^v(i) - \frac{2y(i)}{d_v} \\
&= 1 + \frac{1}{d_v} + \|y\|_2^2 - \frac{2}{d_v} \cdot \|w_b^v\|_1 + 2\langle y, w_b^v \rangle - \frac{2}{d_v} \cdot \|y\|_1 \\
&\leq 1 + \frac{1}{d_v} + \frac{1 - \gamma}{d_v} - \frac{2}{d_v} \cdot \|w_b^v\|_1 + 2\langle y, w_b^v \rangle - \frac{2(1 - \gamma)}{d_v} \\
&= 1 + \frac{1 + 1 - \gamma - 2(1 - \gamma)}{d_v} - \frac{2}{d_v} \cdot \|w_b^v\|_1 + 2\langle y, w_b^v \rangle \\
&= 1 + \frac{\gamma}{d_v} - \frac{2}{d_v} \cdot \|w_b^v\|_1 + 2\langle y, w_b^v \rangle
\end{aligned}$$

Every $y(i)$ can be written as $1/d_v - z(i)$ for some non-negative residual $z(i)$ with $\sum_{i \in S_v} z(i) = \gamma$. So we continue the above equations as follows:

$$\begin{aligned}
\|x + y\|_2^2 &\leq 1 + \frac{\gamma}{d_v} - \frac{2}{d_v} \cdot \|w_b^v\|_1 + 2\langle y, w_b^v \rangle \\
&= 1 + \frac{\gamma}{d_v} - \frac{2}{d_v} \cdot \|w_b^v\|_1 + \frac{2}{d_v} \cdot \|w_b^v\|_1 - 2\sum_{i \in S_v} z(i) w_b^v(i) \\
&= 1 + \frac{\gamma}{d_v} - 2\sum_{i \in S_v} z(i) w_b^v(i) \\
&\leq 1 + \frac{\gamma}{d_v} - 2 \cdot \frac{b_{min}}{\sqrt{d_v}} \cdot \sum_{i \in S_v} z(i) \\
&= 1 - \frac{2b_{min}\gamma}{\sqrt{d_v}} + \frac{\gamma}{d_v}.
\end{aligned}$$

The last inequality holds because Lemma B.4 implies that $w_b^v(i) \geq \frac{b_{min}}{\sqrt{d_v}}$. We conclude that:

$$\|\Delta_{user}\|_2 \leq \sqrt{1 - \frac{2b_{min}\gamma}{\sqrt{d_v}} + \frac{\gamma}{d_v}}.$$

We can now bound the $\ell_2$-sensitivity of the entire algorithm as

$$\|\Delta\|_2 \leq \|\Delta_{reroute}\|_2 + \|\Delta_{user}\|_2 \leq \frac{\alpha\gamma}{\sqrt{d_{max}}} + \sqrt{1 - \frac{2b_{min} \cdot \gamma}{\sqrt{d_v}} + \frac{\gamma}{d_v}}.$$

As the expression $-\frac{2b_{min}}{\sqrt{d_v}} + \frac{1}{d_v}$ is increasing for $d_v \geq \frac{1}{b_{min}^2}$, the right hand side is maximized with $d_v = d_{max}$:

$$\|\Delta\|_2 \leq \frac{\alpha\gamma}{\sqrt{d_{max}}} + \sqrt{1 - \frac{2b_{min} \cdot \gamma}{\sqrt{d_{max}}} + \frac{\gamma}{d_{max}}}.$$

Our goal is to choose $\alpha \in [0, 1]$ such that the right hand side is upper bounded by 1. We note that for $\gamma = 0$, the above inequality proves this desired upper bound. For other cases, $\gamma \in (0, 1]$, it is achieved when,

$$\alpha \leq \frac{\sqrt{d_{max}}}{\gamma} \left(1 - \sqrt{1 - \frac{2b_{min} \cdot \gamma}{\sqrt{d_{max}}} + \frac{\gamma}{d_{max}}}\right). \tag{4}$$

By Lemma B.8 with $C = 2b_{min}$ and using the restrictions $d_{max} > 1$ and $\frac{1}{2} \leq b_{min} \leq 1$, it suffices to choose

$$\alpha = b_{min} - \frac{1}{2\sqrt{d_{max}}}. \tag{5}$$

$\square$

*Proof of Theorem B.1.* Note that both rounds of MAD2R (Algorithm 5) correspond to running the WeightAndThreshold meta-algorithm (Algorithm 1) with privacy parameters $(\varepsilon_1, \delta_1)$ and $(\varepsilon_2, \delta_2)$, respectively. The only difference is that we only materialize $\tilde{w}_1$ rather than the entire vector $\tilde{w}_{ext}$ from the first round. The functionality of our algorithm would be equivalent if we instead materialized the full vector as we only query weights on items in $\mathcal{U}$ and we never output the vector. Therefore, we will invoke Theorem 3.3 twice and apply basic composition to prove the privacy of MAD2R. By Theorem 3.3, it suffices to show that the $\ell_2$ and novel $\ell_\infty$ sensitivities of the weight algorithm MAD are bounded by 1 and $\frac{b_{max}}{\sqrt{t}}$, respectively. This follows directly from Lemma B.3, and Lemma B.6. $\square$

### B.3. Technical Lemma

**Proposition B.7** (Taylor expansion of $\sqrt{1+x}$ as $x \to 0$)**.**

$$\lim_{x \to 0} \sqrt{1+x} = \sum_{n=0}^{\infty} \frac{\prod_{k=1}^{n}\left(\frac{3}{2} - k\right)}{n!} x^n.$$

**Lemma B.8.** *For a constant $1 \leq C \leq 2$, consider the following function of $x$ parameterized by an auxiliary variable $y$:*

$$f(x; y) = \frac{\sqrt{y}}{x} \left(1 - \sqrt{1 - \frac{Cx}{\sqrt{y}} + \frac{x}{y}}\right). \tag{6}$$

*For any $y > 1$,*

$$\inf_{x \in (0,1]} f(x; y) = \frac{C}{2} - \frac{1}{2\sqrt{y}}.$$

*Proof.* To minimize $f$, we will evaluate the function at any stationary points (in terms of $x$) as well as the boundaries $x = 1$ and $x \to 0$. Consider the derivative

$$\frac{d}{dx} f(x; y) = -\frac{\sqrt{y}}{x^2}\left(1 - \sqrt{1 - \frac{Cx}{\sqrt{y}} + \frac{x}{y}}\right) + \frac{\sqrt{y}}{x}\left(\frac{\frac{C}{\sqrt{y}} - \frac{1}{y}}{2\sqrt{1 - \frac{Cx}{\sqrt{y}} + \frac{x}{y}}}\right).$$

Let $A = \sqrt{1 - \frac{Cx}{\sqrt{y}} + \frac{x}{y}}$. To look for stationary points and will set the derivative of $f$ to zero:

$$\frac{d}{dx} f(x; y) = 0 \iff -\frac{1}{x}(1 - A) + \frac{\frac{C}{\sqrt{y}} - \frac{1}{y}}{2A} = 0 \iff x\left(\frac{C}{\sqrt{y}} - \frac{1}{y}\right) < 2A(1 - A)$$

We expand $A^2$ to get the simpler form:

$$\frac{Cx}{\sqrt{y}} - \frac{x}{y} = 2A - 2\left(1 - \frac{Cx}{\sqrt{y}} + \frac{x}{y}\right) \iff 0 = 2A - 2 + \frac{Cx}{\sqrt{y}} - \frac{x}{y} = -A^2 + 2A - 1 = -(A - 1)^2$$

$$\iff A = 1.$$

From the definition of $A$,

$$A = 1 \iff 1 - \frac{Cx}{\sqrt{y}} + \frac{x}{y} = 1 \iff Cx\sqrt{y} = x \iff y = \frac{1}{C^2}.$$

As $C \geq 1$, in the parameter regime $y > 1$, $f$ has no stationary points.

It remains to check the boundary point $x = 1$ and the function $f$ in the limit as $x \to 0$. For $x = 1$, the claim is reduced to this simple inequality:

$$\sqrt{y}\left(1 - \sqrt{1 - \frac{C}{\sqrt{y}} + \frac{1}{y}}\right) \geq \frac{C}{2} - \frac{1}{2\sqrt{y}} \iff 2y\left(1 - \sqrt{1 - \frac{C}{\sqrt{y}} + \frac{1}{y}}\right) \geq C\sqrt{y} - 1$$

$$\iff (2y - C\sqrt{y} + 1)^2 \geq 4y^2\left(1 - \frac{C}{\sqrt{y}} + \frac{1}{y}\right)$$

$$\iff 4y^2 + C^2 y + 1 - 4Cy\sqrt{y} + 4y - 2C\sqrt{y} \geq 4y^2 - 4Cy\sqrt{y} + 4y$$

$$\iff 1 + C^2 y - 2C\sqrt{y} \geq 0$$

$$\iff (C\sqrt{y} - 1)^2 \geq 0$$

which holds for any value of $y$. In the rest of the proof, we focus on the limit as $x \to 0$.

Via the Taylor expansion of Proposition B.7,

$$\lim_{x \to 0} f(x; y) = \lim_{x \to 0} \frac{\sqrt{y}}{x}\left(1 - \sum_{n=0}^{\infty} \frac{\prod_{k=1}^{n}\left(\frac{3}{2} - k\right)}{n!}\left(-\frac{Cx}{\sqrt{y}} + \frac{x}{y}\right)^n\right)$$

$$= \lim_{x \to 0} \frac{\sqrt{y}}{x}\sum_{n=1}^{\infty} -\frac{\prod_{k=1}^{n}\left(\frac{3}{2} - k\right)}{n!}\left(-\frac{Cx}{\sqrt{y}} + \frac{x}{y}\right)^n.$$

Note that the coefficients in the summation are upper bounded in magnitude by 1 as the sequence of terms in the descending factorial in the numerator is dominated by the sequence in the factorial in the denominator. We also note that the absolute value of the coefficient of the first term is a constant, i.e. $\frac{1}{2}$. So, in the limit, the summation is dominated by the lowest order terms with respect to $x$ which correspond to $n = 1$. In this case, the coefficient is $-\frac{1}{2}$ and the limit evaluates to

$$\lim_{x \to 0} f(x; y) = \frac{\sqrt{y}}{x}\left(\frac{Cx}{2\sqrt{y}} - \frac{x}{2y}\right) = \frac{C}{2} - \frac{1}{2\sqrt{y}}.$$

$\square$

## C. Utility

### C.1. Stochastic Dominance Proof

**Theorem C.1.** *Let $\beta \geq 0$ be the parameter controlling the adaptive threshold excess. Let $U$ be the set of items output when using* `Basic` *as the weighting algorithm and let $U^*$ be the set of items output when using unbiased* `MAD` *as the weighting algorithm. Then, for items $i \in \mathcal{U}$,*

- *If $\Pr(i \in U) < \Phi(\beta)$, then $\Pr(i \in U^*) \geq \Pr(i \in U)$.*

- *Otherwise, $\Pr(i \in U^*) \geq \Phi(\beta)$.*

*Proof of Theorem C.1.* Let $w$ and $w^*$ be the weights produced by `Basic` and `MAD`, respectively. Let $I_{adapt} = \{u \in [n] : |S_u| \leq d_{max}\}$ be the items which participate in adaptive rerouting. For any item $i \in \mathcal{U}$, we will consider it's initial weight under the adaptive algorithm:

$$w^*_{init}(i) = \sum_{u \in I_{adapt}: i \in S_u} \frac{1}{|S_u|}.$$

We will proceed by cases.

**Case 1:** $w^*_{init}(i) \leq \tau$. In this case, $w^*(i) \geq w(i)$. In the adaptive algorithm, no weight is truncated from the initial weights, and so each user contributes to the final weight of an item $\frac{1}{\sqrt{|S_u|}}$ plus rerouted weight from other items. As the weight on item $i$ only increases for the adaptive algorithm compared to the basic algorithm, the probability of outputting $i$ also can only increase.

**Case 2:** $w^*_{init}(i) > \tau$. In this case in the adaptive algorithm, the initial weight is truncated to $\tau$, excess weight is rerouted, and a final addition of $\frac{1}{\sqrt{|S_u|}} - \frac{1}{|S_u|}$ is added. As $|S_u| \geq 1$, the final weight $w^*(i) \geq \tau$. Then, $i \in U^*$ if the added Gaussian noise does not drops the weight below the threshold $\rho$, i.e., if the noise is greater than or equal to

$$\rho - \tau = \rho - (\rho + \beta\sigma) = -\beta\sigma.$$

As the noise has zero mean and standard deviation $\sigma$, this probability is exactly $1 - \Phi(-\beta) = \Phi(\beta)$. $\square$

### C.2. Example showing a gap between `MAD` and `Basic` DP SIPS

While Theorem C.1 bounds the worst-case behavior of our algorithm compared to the basic algorithm, as `MAD` increases the weight of items below $\tau$ compared to the basic algorithm, it will often have a larger output. We show a simple, explicit example where our algorithm will substantially increase the output probability of all but one item.

Here, we demonstrate an explicit setting where `MAD` outperforms the baselines. There are $n$ users each with degree 3 as well as a single heavy item $i^*$ and $m$ light items. Each user's set is comprised of $i^*$ as well as two random light items. Under the basic algorithm, each user will contribute $1/\sqrt{3}$ to each of their items. Therefore, the weights under `Basic` are

$$w(i) = \begin{cases} \frac{n}{\sqrt{3}} & \text{if } i = i^* \\ \frac{2n}{\sqrt{3}m} < \frac{1.16n}{m} & o.w. \end{cases}.$$

On the other hand, assuming $n >> \tau$, `MAD` will reroute almost all of the initial weight on the heavy item back to the users, so each user will have excess weight approximately $1/3$. For $d_{max} = 3$, we get discount factor $\alpha > 0.5$. So, each user will send approximately $1/18$ weight to each of its items. The weights under `MAD` are

$$w(i) = \begin{cases} \tau + \frac{n}{18} & \text{if } i = i^* \\ \frac{2n}{m}\left(\frac{1}{\sqrt{3}} + \frac{1}{18}\right) < \frac{1.27n}{m} & o.w. \end{cases}.$$

In this setting, our algorithm will assign close to $10\%$ more weight to the light items (resulting in substantially higher probability of output) compared to the basic algorithm. If $n/m$ is close to the true threshold $\rho$, this gap will have a large

effect on the final output size. We empirically validate this for $n = 15{,}000, m = 1000, \varepsilon = 1, \delta = 10^{-5}$. Our algorithm returns 610 items on average. The basic algorithm returns 519 items while DP-SIPS with a privacy split of $5\%, 15\%, 80\%$ or $10\%, 90\%$ returns 332 or 514 items, respectively. In all cases, as expected, our algorithm has significantly higher average output size.

# D. Additional Figures

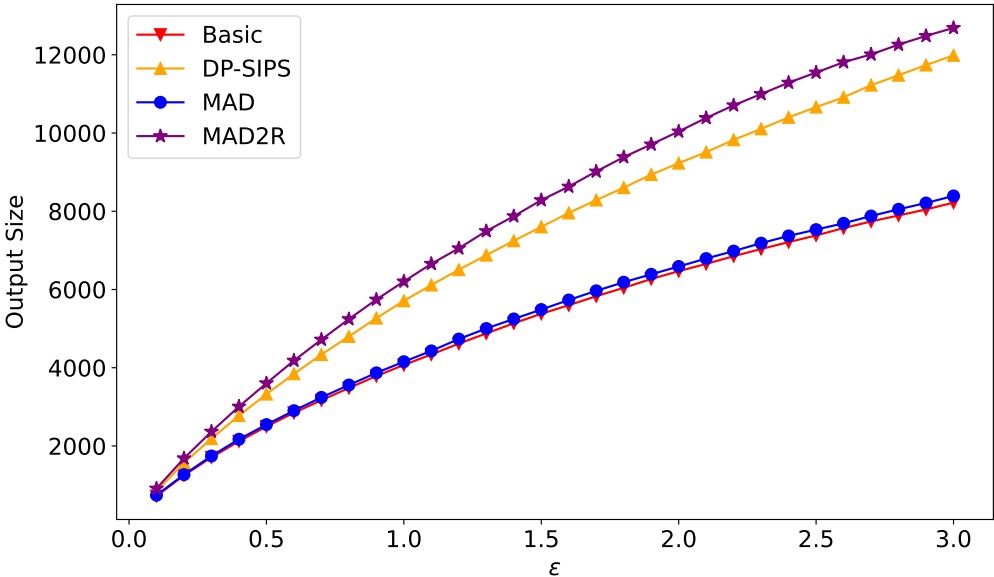

Figure 2: Comparison of output size across parallel algorithms while varying privacy parameter $\varepsilon$ on the Reddit dataset. Other parameters are fixed as described in Section 5 with a fixed privacy split of $[0.1, 0.9]$ for DP-SIPS and MAD2R. The relative performance of algorithms does not change with this parameter. Increasing $\varepsilon$ significantly improves performance at the cost of privacy by lowering the required noise and threshold.

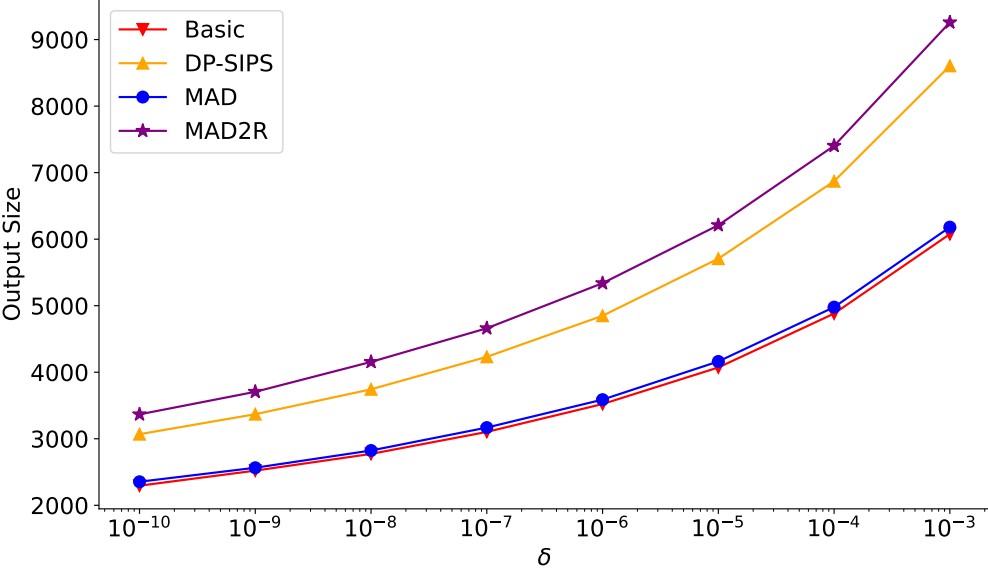

Figure 3: Comparison of output size across parallel algorithms while varying privacy parameter $\delta$ on a log-scale on the Reddit dataset. Other parameters are fixed as described in Section 5 with a fixed privacy split of $[0.1, 0.9]$ for DP-SIPS and MAD2R. The relative performance of algorithms does not change with this parameter. Increasing $\delta$ significantly improves performance at the cost of privacy by lowering the required noise and threshold.

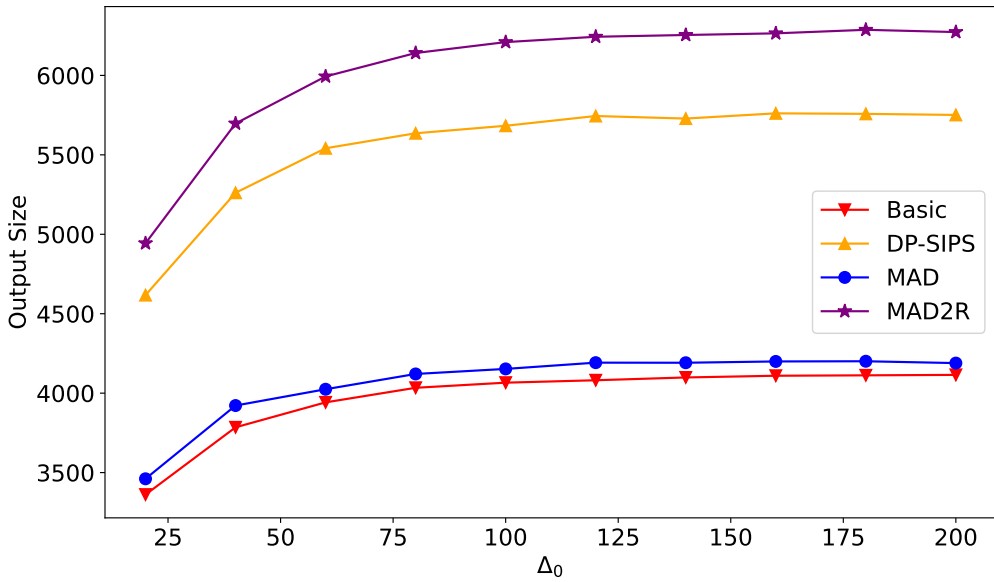

Figure 4: Comparison of output size across parallel algorithms while varying maximum set size parameter $\delta_0$ on the Reddit dataset. Other parameters are fixed as described in Section 5 with a fixed privacy split of $[0.1, 0.9]$ for DP-SIPS and `MAD2R`. The relative performance of algorithms does not change with this parameter, and good results are achieved as long as it is not too small.

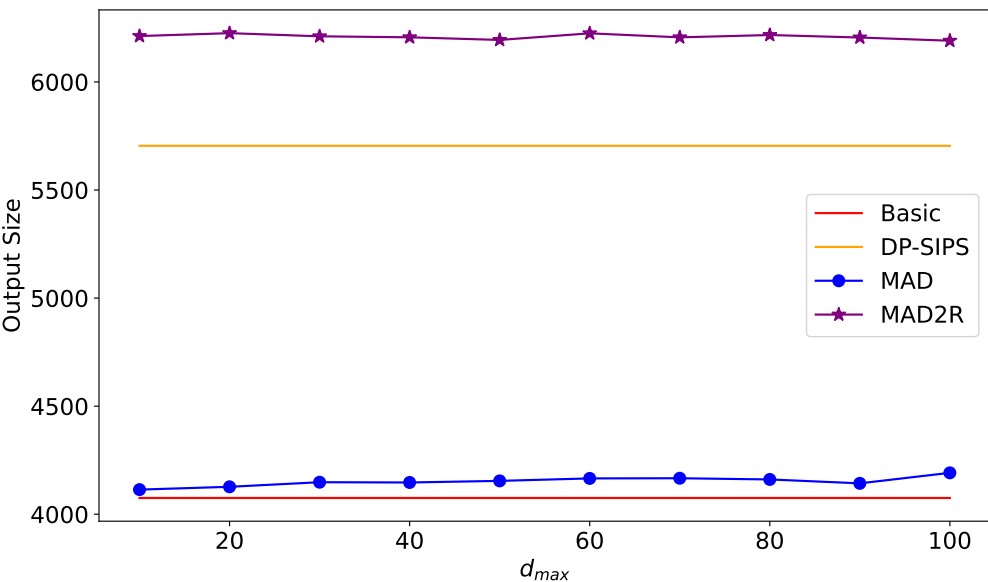

Figure 5: Comparison of output size across parallel algorithms while varying the parameter $d_{max}$ of our algorithms on the Reddit dataset. As this parameter is only used by `MAD` and `MAD2R`, the performance of the baselines is fixed. Other parameters are fixed as described in Section 5 with a fixed privacy split of $[0.1, 0.9]$ for DP-SIPS and `MAD2R`. The performance of our algorithm is relatively insensitive to this parameter.

