# OpenReview forum: "Scalable Private Partition Selection via Adaptive Weighting"
_ICML.cc/2025/Conference — ICML 2025 poster_

### Official Review · Reviewer_bvMj · 2025-02-17

**Overall Recommendation:** 4

**Summary:**

This work focuses on differentially private partition selection problem, which is very crucial for many underlying applications like NLP. In this problem, the system would like to extract the data as much as possible it can do from users while preserving users’ privacy. It proposes an adaptive and non-uniform weighting algorithm (MAD) with ability of parallelization to tackle the problem. In addition, it also proposes MAD2R to maximize the output in the second round of mad.

**Claims And Evidence:**

Claims made in this work are well expresses in the Algorithm 2 and 5 for MAD and MAD2R respectively. With the purple statements in those algorithms, each step is clarified with easy understanding.

**Essential References Not Discussed:**

To my best knowledge, authors covered most essential works related to this topic.

**Experimental Designs Or Analyses:**

The experimental designs in this work is great because it first compares with sequential algorithms like PolicyGaussian in Gopi et al., 2020 and GreedyUpdate in Carvalho et al., 2022, and then with parallel approaches like DP-SIPS and proposed algorithms in this work. All experimental settings are well explained. If authors can put some figures from Section E in the appendix into Section 5, it would provide a better visual sense to audiences.

**Methods And Evaluation Criteria:**

MAD and MAD2R are clear with detailed pseudo codes, which are easy to reproduce for the future research and datasets selected are diverse from small scale to large size to demonstrate scalability and parallelization of this work.

**Other Comments Or Suggestions:**

Can you provide a comprehensive conclusion section by concluding the work and also providing some future work according to private partition selection?

**Other Strengths And Weaknesses:**

Strengths:

1. Compared to prior works, this work can be better to allocate weights instead of using same noise and threshold for a certain privacy parameters.

Weakness:

1. As mentioned in the previous sections, authors put too many details in the appendix and point audience to read everything back and forth. The reading experience will lower down by this kind of management.

**Questions For Authors:**

Why does users with a too small or large cardinality use non-adaptive approach in MAD?

**Relation To Broader Scientific Literature:**

1. Korolova et al., 2009 pioneered the DP partition selection problem by constructing the sampling-and-weighting format, which is the closest works for the problem on massively parallel computation, but they are based on uniform weighting strategy, which is considered loss in privacy.
2. Prior algorithms like Gopi et al., 2020 and Carvalho et al., 2022 only run on a single machine storing data in the memory while this work can efficiently work for massively parallel computation systems with constant rounds.
3. Swanberg et al., 2023 is the most related work to this work. DP-SIPS in that paper is the pioneering work with weighting other than basic weighting and a parallel setting. This work also compares with DP-SIPS deeply in MAD2R section.

**Theoretical Claims:**

The theory mentioned in the paper looks convincing (with reading appendix), but authors hide too many details in the appendix, such as proofs to the theorems. I think authors can shorten the narrative parts in the contribution section and put add some more details for Theorem 1.1 and 1.2 instead of saying they are informal version of the things they discuss in the appendix.

---

> ### Author Rebuttal · Authors · 2025-03-31
>
> Thank you for taking the time to review our paper! We respond to your comments and questions below.
>
> ### Organization with the appendix and conclusion section
>
> Thanks for the comment and suggestions! We agree that it would be a preferable reading experience to bring more content from the appendix into the main body but are limited by the page limits. We will address your specific suggestions in terms of what content to move into the main body in the final paper. Likewise, we will include a conclusion in the final paper.
>
> ### Reason for limit on adaptive degree (from Questions)
>
> This is a good question: the reasons for the upper and lower degree bounds for which users can be adaptive are to ensure that the sensitivity of MAD is bounded, and we elaborate on the technical reasons below.
>
> Upper bound: In the initial $\ell_1$ bounded allocation of MAD, we limit the adaptive users to rerouting weight inversely proportional to $1/d_{\max}$ rather than $1/d$. If we allowed users with degree more than $d_{\max}$ to be adaptive, this allocation would have an $\ell_1$ norm more than $1$. The choice to use $1/d_{\max}$ weights is used throughout the proof of Lemma B.6 regarding the $\ell_2$ sensitivity of MAD. Without this restriction, the algorithm would not be private as a novel high degree user could cause many low degree users to reroute their weight which could be overly concentrated on a few items. By using $1/d_{\max}$, we limit the amount low degree users can reroute to any given item.
>
> Lower bound: The lower bound on the degree of adaptive users is used on line 1001 in order to ensure that the final $\ell_2$ sensitivity of MAD can be bounded by 1. For users with too small degrees, we cannot bound the expression at the end of Lemma B.6.

---

### Official Review · Reviewer_9UnS · 2025-03-11

**Overall Recommendation:** 3

**Summary:**

This paper studies private partition selection (or set union) under the constraint of differential privacy (DP). Here, they consider user-level DP (UDP), where each user is allowed to contribute multiple points to the dataset. It provides both theoretical and experimental results to to show that their work improves on the prior work. Also, there is more computational efficiency in this work owing to being able to parallelize the algorithm. This is the first paper that is able to achieve both utility and the ability to be parallelized at the same time. They test on 9 different datasets, including on those used in recent prior work. It provides two algorithm, $MAD$ and $MAD2R$, where the latter is a multi-round version of the former, and provides better guarantees, as well. The techniques rely on weighting and reweighting the items privately in order to recover items closer to the tail ends of the dataset.

**Claims And Evidence:**

The claims that are made in the paper are as follows.
1. Their work satisfies $(\varepsilon,\delta)$-DP.
2. Their algorithms show statistical dominance over the prior work.
3. Their work can be run on a distributed system whilst being parallelized.
4. Their work also performs better empirically on different datasets than the prior work.

**Essential References Not Discussed:**

None that I can think of.

**Experimental Designs Or Analyses:**

As said above, the experimental design was simple and relevant for this problem. The conclusions drawn from the experiments were sound. I checked the plots in the appendix, in addition to the table in the main body.

**Methods And Evaluation Criteria:**

They do. The experimental results compare the right form of utility with that of the prior work for different datasets and privacy parameters.

**Other Comments Or Suggestions:**

Nothing major. I would address my comment in Weakness 2 above in the paper, for sure. Also, indicating how many items that were missed were actually at the tail end of the dataset, and why they were missed by the algorithm.

**Other Strengths And Weaknesses:**

Strengths:
1. This is the first paper that is both accurate and computationally efficient in the sense that the algorithm could be parallelized. This is a big win in terms of practicality.
2. The utility dominates that of the prior works, including on the datasets that were being previously used to test those relevant algorithms.
3. They have used 9 datasets to test their algorithms, which is comprehensive enough.

Weaknesses:
1. Whilst their results have improved on the prior work, often the percentage of improvement isn't that high, especially given that the prior works weren't super high utility either. It's good though that their work is consistently better than all the prior work for all the datasets.
2. One more thing that I'm not entirely sure about is the absolute utility. I'm referring to the percentage of the items recovered by their algorithms. For example, the Common Crawl dataset has over 1.8 billion items, but you're managing to recover only about 29 million. Whilst that is an improvement over the prior work, it still doesn't get to recovering even 50% of the total items.

**Questions For Authors:**

1. Is my assessment in Weakness 2 correct? If so, what was the reason so many items were missed?
2. Can we get even more utility by having more rounds for the very low-weight items? In DP statistics, often multi-round algorithms are used, for example, for Gaussian covariance estimation by Kamath et al 2019 and Kamath et al 2020. Here, we could repeatedly remove heavy items, and look at lighted items in the next round. Could that kind of an idea help? Maybe it won't just be post-processing though.

**Relation To Broader Scientific Literature:**

This is a very relevant problem in ML. This can fit as a building block for privacy-preserving ML applications. This certainly improves significantly on the prior work, but the problem by itself wouldn't be as interesting, if it didn't have applications in other areas. All in all, it does help in larger applications, so I don't have any complaints with the problem being solved and the practicality of their solutions.

**Theoretical Claims:**

I vaguely skimmed through the proofs as they were all part of the appendix, as opposed to the main body. They looked alright in that read.

---

> ### Author Rebuttal · Authors · 2025-03-31
>
> Thank you for taking the time to review our paper!
>
> ### Absolute utility (Question 1)
>
> Your intuition that the raw number of items output by our algorithm (and any private algorithm) is affected by the long tails of the dataset is absolutely correct! We report some interesting new results on how the frequency of the items in the raw data affect their output probability. We will include these in our paper.
>
> It is widely-known that in many user-provided datasets, items have a heavy-tailed distribution where many items are infrequent and some items are highly prevalent. It is a requirement of any DP algorithm that **items belonging to few users cannot be output**. Thus, privacy can prevent outputting the majority of items in heavy-tailed datasets. On the other hand, while fewer in number, the more frequent items are often the most important, especially for downstream applications. Hence, it is essential to output as many items as possible among the ones that are possible to be output with privacy protections.
>
> We demonstrate that MAD2R is doing a good job of covering high frequency items that can actually be safely output, showing that most of the loss comes from items that (essentially) cannot be output due to privacy by any private algorithm. Below, we analyze the Reddit dataset and the very large Common Crawl dataset.
>
> **Reddit**
>
> The Reddit dataset has 143,556 unique items, of which 83,313 (58%) are singleton items–they only appear in a single user’s set. Any algorithm which outputs any of these singleton items is *not private* by definition, as it is leaking private information belonging to a single user. For any private algorithm with acceptable privacy settings, outputting items with very small frequencies is also simply not possible.
>
> MAD2R outputs 6,340 items (4.4%) on the Reddit dataset with the standard parameter settings in our paper. On the other hand, 98% of users have at least one item output by MAD2R. And 45% of the entries (user-item pairs) belong to an item which is output by MAD2R.
>
> We break down the number of items of different frequencies which our algorithm returns. Notice that MAD2R outputs the vast majority of items with large enough frequency (as expected, very few low frequency items are output and no singleton items are output). The numerator is the number of such items returned by MAD2R and the denominator is the number of items in the dataset in the frequency range:
>
> Fraction of items returned with frequency in [1,2): 0/83313 = 0.0000
>
> … in [2,50): 28/134765 = 0.0002
>
> … in [50,100): 878/3207 = 0.2738
>
> … in [100,200): 2007/2274 = 0.8826
>
> … in [200,500): 1668/1680 = 0.9929
>
> … in [500,infinity): 1630/1630 = 1.0000
>
> **Common Crawl**
>
> The results for Common Crawl are very similar to the ones in Reddit. MAD2R outputs 28,967,624 out of 1,803,720,630 (distinct) items. That is 1.6% of the total. However, as we can see in the table below 61% of all items are unique (and can’t be output by any private algorithm). Moreover, our algorithm covers the majority of items that appear in at least 200 users (out of billions of users). More importantly, 99.9% of users are covered by at least one item and 98+% of entries in the dataset belong to an item in output by our algorithm.
>
> Fraction of items returned with frequency in [1,100000000000): 28967624/1803720630 = 0.016
>
> … in [1,2): 21/1109806713 = 0.000
>
> … in [2,50): 251364/627294180 = 0.000
>
> … in [50,100): 1445285/21404804 = 0.068
>
> … in [100,200): 4100448/14714470 = 0.279
>
> … in [200,500): 7366701/12902081 = 0.571
>
> … in [500,1000): 5105166/6326548 = 0.807
>
> … in [1000,100000000): 10697894/11271089 = 0.949
>
> This suggests that the utility is very high for downstream applications and a large fraction of the loss is on items that should not be output by a private algorithm (as they are unique to a small number of users).
>
> Thank you for suggesting this analysis. We will add more details about this in the final paper.
>
> ### More rounds for low-weight items (Question 2)
>
> As you suggest, spending more resources on moderate weight items is the key challenge to improve performance for this problem (we say moderate as opposed to low as very low weight items simply cannot be output while maintaining privacy). However, there are two challenges with the general concept of spending more rounds on moderate weight items.
>
> First, we do not know what items are going to have moderate weight apriori. This is why our contribution of using noisy weights from prior rounds is so useful and opens up a new design space for parallel algorithms. Second, using more rounds does not necessarily translate to better performance. There is a fixed privacy budget and more rounds means less budget per round and therefore more noise and higher thresholds.  For MAD2R, we choose 2 rounds in order to maximize the privacy budget in each round to get accurate noisy estimates from the first round and then to minimize the threshold in the second round when we make use of those noisy estimates.

---

### Official Review · Reviewer_Jbgz · 2025-03-12

**Overall Recommendation:** 4

**Summary:**

The paper deals with the problem of differentially private set union.
Each user holds a subset of items from an unbounded universe, and the goal is to output as many items from the union of all the users' sets while maintaining user-level privacy.
Useful intuition for the problem is that items held by a single user should not be output, whereas items held by many users are fine to output, motivating weighting-based algorithms.
Such algorithms work in the following way: 1) each user contributes a weight vector over the items they hold, 2) all weights are summed together centrally, 3) each item is privately checked to be above a threshold, 4) item above the threshold are released.

The processes in which these weights are derived from users' sets distinguishes these algorithms.
One approach is to greedily produce weights by sequentially interacting with each user and letting their weight update depend on every past weight update (Gopi et al., 2020; Carvalho et al., 2022).
This approach produces good results, but is not apt for massively parallel computation.

A more scalable approach is to interact with each user in rounds.
The first such scalable algorithm was uniform weighting (Korolova et al., 2009) where each user's items contribute uniformly to the weight, and this was performed for one round.
Later DP-SIPS (Swanberg et al., 2023) made the amendment of splitting the privacy budget over multiple rounds, and running uniform weighting in each round.
Items selected in preceding rounds got removed in future rounds, and finally the union of the private unions from each individual round is released.

This paper improves on the uniform weighting procedure.
Simplified, they show that it is possible to make it adaptive (Algorithm 2, MAD).
After running uniform weighting, they truncate weights $w(i) \geq \tau$, where $\tau$ is a parameter, to $\tau$, and re-route these weights back to users, proportionally to how much they contributed.
The intuition is that items with large weights are likely to exceed the private threshold check, so by re-allocating weight to other items (if that is possible to do at no cost in privacy) we can expect to be able to release more items.
The authors show that, indeed, this re-routing is possible to perform without increasing the sensitivity of the procedure (Lemma 4.1 and Lemma 4.2), and that it stochastically dominates uniform weighting (Theorem 1.2).

They also consider a biased 2-round version of MAD (Algorithm 5, MAD2R), where the weights of items not released in the first round are re-used for the second round.
Essentially, items with very low weights in the first round are removed from the second round (because they are unlikely to be output, and so the weight can be better spent elsewhere), and items with very large weight are biased to not overshoot the threshold by as much (also allowing for shifting weights around).

The paper supports the theoretical results with empirical work (Section 5, Table 1).
The authors demonstrate that their single and 2-round algorithms (MAD and MAD2R) improve on existing parallel algorithms for all shown datasets.
Sequential algorithms consistently outperform the parallel algorithms, which makes sense and is well-explained.

## update after rebuttal

I find that the authors have adequately addressed questions raised by the reviewers, and I appreciate the additional experiments they have run in support of their rebuttal. For my own questions, the privacy budget split for DP-SIPS and the source of the improvement for MAD2R is now clearer to me. I have upgraded my score.

**Claims And Evidence:**

Yes.

**Essential References Not Discussed:**

None that I am aware of.

**Experimental Designs Or Analyses:**

Yes, the experiments look sound to me.

**Methods And Evaluation Criteria:**

Yes.

**Other Comments Or Suggestions:**

Typos:
363r: adpativity

**Other Strengths And Weaknesses:**

Strengths:
1. Well-written paper with good structure.
2. The idea of shifting around weight for items to increase the chance of more items exceeding the threshold is natural and makes sense. The fact that it is possible to do without increasing the sensitivity of the weighting procedure is surprising to me, and nice.
3. The stochastic dominance of MAD over Basic is a nice property of the paper.
4. The empirical work supports the paper well.

Weaknesses:
1. The choice of number of rounds in the experiments for DP-SIPS, and only considering a 2-round version of MAD, seem like arbitrary choices.
2. The improvement MAD2R achieves over DP-SIPS is not that large.
3. From the experiment, it is unclear what drives the improvement, and it is not easily inferred from the paper.
In particular, the improvement MAD2R achieves over DP-SIPS is not obviously as much due to MAD, as much as it is from re-using the weights across rounds.

**Questions For Authors:**

I ask the following questions to better understand the empirical results in Section 5.

Questions:
1. Can you comment on what you believe drives the improvement shown in Table 1? Since MAD slightly improves over Basic, it could be that the improvement MAD2R achieves over DP-SIPS comes largely from using weights from the first round in the second round, rather than the improvement "within" each round, i.e. MAD's improvement over uniform weighting.
2. If I understand it correctly, MAD2R with $d_{max}=0$ is essentially running 2-round DP-SIPS and re-uses weights in the second round. Did you try running DP-SIPS combined with re-using the weights across rounds?
3. Is there any reason why you only consider a 2-round version of MAD (and why DP-SIPS only goes up to 3 rounds)?

**Relation To Broader Scientific Literature:**

The paper improves on existing parallel algorithms for parallel private partition selection (Korolova et al., 2009; Swanberg et al., 2023) and compares against sequential algorithms for the same problem (Gopi et al., 2020; Carvalho et al. al., 2022).
I am not familiar with other related work in the literature that warrant mentioning in this context.

**Theoretical Claims:**

The intuition conveyed in the main paper is mostly clear, and I checked that proofs existed in the appendix, but I did not check any proofs in detail.

---

> ### Author Rebuttal · Authors · 2025-03-31
>
> Thank you for taking the time to review our paper! We respond to your comments and questions below.
>
> ### Number of rounds for MAD2R and DP SIPS (Question 3)
>
> For this problem, using more rounds does not necessarily translate to better performance. There is a fixed privacy budget and more rounds means less budget per round and therefore more noise and higher thresholds. We address the specific question of more rounds for MAD2R and DP SIPS separately below.
>
> **DP SIPS:**
>
> In our paper, we report the best performance of DP SIPS with either 2 or 3 rounds (also with 1 round as then DP SIPS is equivalent to Basic). We do this in order to give this baseline a more favorable comparison compared to our results (notice that we allow DP SIPS to non-privately use the best parameter setting, something we do not do for our algorithms). We used 2 or 3 rounds for DP SIPS following the suggestions of the DP SIPS paper which uses three rounds as a standard choice. In our experimentation, we confirm  that 2 or 3 rounds of DP SIPS with a good privacy split (such as [0.05, 0.15, 0.8]) generally performs well without seeing benefits from more rounds. Below, we report many choices of privacy splits for 4 and 5 rounds on the Reddit dataset for context confirming that no improvement is observed. We report the number of rounds, the privacy split across rounds, and the output size. The best result is achieved in 3 rounds and the second best at 2 rounds.
>
> 1 round, [1.0]: 4059
>
> 2 rounds, [0.10, 0.90]: 5731
>
> **3 rounds, [0.05, 0.15, 0.80]: 5797**
>
> 4 rounds, [0.10, 0.10, 0.20, 0.60]: 5037
>
> 4 rounds, [0.10, 0.10, 0.10, 0.70]: 5199
>
> 4 rounds, [0.05, 0.05, 0.05, 0.85]: 5404
>
> 4 rounds, [0.05, 0.10, 0.15, 0.70]: 5423
>
> 4 rounds, [0.01, 0.04, 0.05, .90]: 5598
>
> 5 rounds, [0.05, 0.10, 0.15, 0.20, 0.50]: 4452
>
> 5 rounds, [0.10, 0.10, 0.10, 0.10, 0.60]: 4666
>
> 5 rounds, [0.05, 0.05, 0.05, 0.05, 0.80]: 5239
>
> **MAD2R:**
>
> We designed MAD2R to utilize two rounds. From any given round, MAD2R is able to use strictly more information than DP SIPS. DP SIPS only gets binary information about whether noisy item weights from the prior round were extremely large (enough to pass a very high threshold). MAD2R uses **all** of the noisy item weights from the prior round. In order to optimize this information, we want to minimize the noise in prior rounds. We choose 2 rounds in order to maximize the privacy budget in each round to get accurate noisy estimates from the first round and then to minimize the threshold in the second round when we make use of those noisy estimates.
>
> An intuition as to why 3 rounds is a good choice for DP SIPS is that the first round captures very frequent elements, the second round captures the next tier of quite frequent elements, and the third round captures remaining moderate frequency elements. Because of our innovation of using noisy weights from prior rounds, we can achieve better results in two rounds by getting and utilizing all of the information about frequent/moderately frequent elements in one shot.
>
> ### Breaking down the performance of MAD2R (Questions 1 and 2)
>
> Thanks for the suggestion of running MAD2R with $d_{\max}=0$ to understand the impacts of adaptivity via MAD vs. adaptivity via reusing noisy weights. We stress that both of these uses of adaptivity are key innovations in this work. The ability to re-use weights from prior rounds is a technical innovation of our work which was not present in prior work and thus should be considered a contribution of our paper. The conceptual idea of reusing weights and the proof that this is private are important parts of our work and a key enabler of additional performance gains.
>
> We ran the numbers for the Reddit dataset and get the following result:
>
> DP SIPS: 5784
>
> **MAD2R: 6215**
>
> MAD2R ($d_{\max}=0$): 6197
>
> Our contribution of using noisy weights from prior rounds to bias MAD2R in the second round constitutes the majority of the improvement over DP SIPS. Utilizing MAD with $d_{\max} > 0$ has added benefit at no cost (as using MAD with adaptivity stochastically dominates the Basic algorithm). On the same dataset, for single round algorithms, MAD beats Basic by 100 items. (for the larger common crawl datasets we output 100,000+ more items). The main effect of MAD within the two round MAD2R is in the first round where we know nothing about the distribution of item weights. In the second round, because we are biasing weights, it is unlikely that many item weights will be significantly overallocated and therefore the amount of wasted weight to reroute is less. Though we only use a 10% privacy budget in the first round, utilizing MAD2R with $d_{\max}=50$ still contributes more than 10% of the benefit of MAD over Basic.

---

### Official Review · Reviewer_jHbQ · 2025-03-17

**Overall Recommendation:** 3

**Summary:**

This work studies the problem of private partition selection, giving two new, highly parallelizable algorithms for privately outputting as large a subset of the union of user datasets as possible under the constraints of user-level privacy (MAD and MAD2R). They prove that their algorithms satisfies approximate differential privacy and argue utility through a “stochastic dominance” theorem, showing that
any elements of the union output by a simple reference algorithm with probability greater than some threshold will also be output with good (but possibly lower) probability by MAD
Any elements of the union output by the simple reference algorithm with probability below that threshold will be output by MAD with probability at least as high.

They empirically evaluate the performance of their algorithms on Higgs, IMDb, Reddit, Finance, Wiki, Twitter, Amazon, Clueweb, and Common Crawl datasets, comparing their utility to similarly parallelizable baselines as well as sequential (but higher utility) algorithms.

**Claims And Evidence:**

The claims made in the paper are primarily supported via proof, with the exception of claims of improvement in utility over an alternative parallel algorithm (DP-SIPS). This is evaluated empirically, and through a constructed example on which MAD can be argued to have higher utility.

**Essential References Not Discussed:**

To the best of my knowledge there are no references that are not discussed.

**Experimental Designs Or Analyses:**

I did not check the validity of the experimental design.

**Methods And Evaluation Criteria:**

Yes, the benchmarks make sense for this application, to the best of my knowledge.

**Other Comments Or Suggestions:**

Page 2: "And outputting"

Page 4. Sigma is used in the pseudocode before it's defined in the text.

Page 6. "For parameters $C_{lb}$, $C_{ub} \geq 0$, Let"

Page 20. "To look for stationary points and will set the derivative of f to zero"

**Other Strengths And Weaknesses:**

Strengths: the paper is well-organized and comparison to prior work is clear. The evaluation demonstrates that their proposed approach to private partition selection may yield improved utility compared to existing parallel algorithms.

Weaknesses: while the authors describe the new algorithmic approach in some detail, the description is largely in terms of parameters of the algorithm and manipulation of weights. I would have benefited from some additional intuition for what these parameters are "doing" (e.g. what is the discount factor, why is it set the way it is?).

In addition, the lack of theoretical justification for utility improvement compared to DP-SIPS makes it hard to assess the significance of the contribution. None of the parallel algorithms, MAD and MAD2R included, improve upon the sequential algorithms, and within the class of parallel algorithms, DP-SIPS outperforms MAD and seems competitive with MAD2R on many datasets. MAD2R does appear to outperform DP-SIPS on average, but I think a more thorough analysis of when a practitioner may want to use MAD2R over DP-SIPS and vice versa would help to contextualize the results.

**Questions For Authors:**

See comments in strengths and weaknesses. What guidance would you offer practitioners on when to use your algorithms as opposed to others?

**Relation To Broader Scientific Literature:**

The paper is very clear about how its contributions relate to prior work. The authors identify two approaches to partition selection in prior work: parallel and sequential algorithms, and compare their proposed algorithms to prior algorithms in both categories, focusing on parallel algorithms (as their new algorithms are parallelizable).

**Theoretical Claims:**

I read the proofs of privacy and utility contained in the appendices and found no issues.

---

> ### Author Rebuttal · Authors · 2025-03-31
>
> Thank you for taking the time to review our paper!
>
> ### When to use MAD2R vs. DP SIPS (from Questions) and theoretical justification compared to DP SIPS
>
> **MAD2R should always be preferred to DP SIPS** by practitioners as it is strictly better (both for fundamental theoretical reasons and confirmed in experiments). Below, we emphasize the justifications in the paper that we provide for this claim.
>
> **Theoretical justification**:
>
> In the design of MAD2R, we made two key independent contributions: (1) the novel single round adaptive algorithm MAD and (2) the ability to reuse noisy weights from previous rounds in a private way to boost utility. The main theorems of our paper prove that both of these design choices preserve $(\varepsilon, \delta)$-DP. Furthermore, we show that MAD stochastically dominates the Basic algorithm (which is the one used in DP SIPS in each step). Directly comparing MAD2R and DP SIPS, since MAD2R uses MAD in the first round and DP SIPS uses Basic, MAD2R will provably outperform DP SIPS in the first round. In the second round, MAD2R is also strictly superior to DP SIPS in an information theoretic sense: thanks to our privacy proof, our algorithm has access to and utilizes strictly more information from the private data than DP SIPS, all with *no* privacy loss. DP SIPS only gets binary information about whether noisy item weights from the prior round were extremely large (i.e., above a very high threshold). MAD2R uses **all** of the noisy item weights from the prior round (not just whether they are above threshold). To summarize, MAD2R is designed to always improve upon DP SIPS by being more adaptive in two ways while maintaining the same privacy guarantees. Doing some without affecting the privacy analysis (and for instance requiring more noise) is a novel contribution of our work.
>
> For further theoretical justification, in Appendix C.2, we show a constructive  example where MAD (even with just one round) will outperform DP SIPS. The theoretical intuition is that if users has a popular item (present in all users) and a normal item (present in only a fraction of the users), DP SIPS will use the same fixed amount of budget in its first round on the popular item and the normal items even though the weight of the popular item may far exceed the threshold. On the other hand, MAD will adaptively reroute weight from the popular item to the normal items within a single round, thus outperforming Basic and DP SIPS. Heterogeneous item frequencies appear in real data (see response to reviewer 9UnS) where we observe both very popular items and many less popular items (i.e., data has a heavy tail distribution).
>
>
> **Empirical justification**:
>
> We verify this claim experimentally by showing that MAD2R outperforms DP SIPS in **every case** we test. It outperforms DP SIPS for each of the 9 datasets and for every parameter setting of $\varepsilon$, $\delta$, $\Delta_0$, and $d_{\max}$.
>
> ### More intuition about parameters
>
> Thanks for the suggestion to provide more intuition about parameters. Here, we discuss some key parameters of our algorithm, how they affect privacy and utility, and how we chose to set them.
>
> Maximum bias $b_{\max}$: A larger maximum bias allows for the algorithm to more aggressively increase weights in the second round to items which didn’t receive very large noisy weights in the first round. It directly affects the novel $\ell_\infty$ sensitivity of the algorithm, thus increasing the threshold. Consider a neighboring dataset with a novel user with some items which don’t appear in any of the existing users’ sets. In the worst-case scenario, this novel user can assign up to $b_{\max}/\sqrt{d}$ weight to any of its novel items.
>
> Minimum bias $b_{\min}$: A smaller minimum bias allows for the algorithm to more aggressively decrease weights in the second round on items which received very large noisy weights in the first round. In order to prove that MAD maintains privacy, we only do adaptive rerouting for users with degrees at least $1/b_{\min}^2$. See the response to Reviewer bvMj for more details.
>
> Maximum adaptive degree $d_{\max}$: Raising the maximum adaptive degree increases the set of users which can participate in adaptive rerouting in MAD. On the other hand, it reduces the amount of weight which is rerouted for each of those users. It is very challenging to design an algorithm which maintains privacy while being adaptive. To do so, we limit the adaptive users to rerouting weight inversely proportional to $1/d_{\max}$ rather than $1/d$. See the response to Reviewer bvMj for more details.
>
> Discount factor $\alpha$: The reason for our choice of the discount is simply to satisfy a technical condition. We set the discount as a variable and perform the $\ell_2$ sensitivity analysis in Lemma B.6. At the end of the analysis, we have a bound on the sensitivity involving this variable (see lines 1005-1020), and we set the variable to be as large as possible so that the upper bound is 1.

---

> > ### Comment · Reviewer_jHbQ · 2025-04-07
> >
> > I thank the authors for their thorough explanation of the relative merits of their algorithm and for providing additional intuition regarding the parameters! I've updated my score accordingly.

---

> > > ### Author Response · Authors · 2025-04-07
> > >
> > > Thank you very much!

---

### Decision · Program_Chairs · 2025-05-01

**Decision:**

Accept (poster)

**Comment:**

The paper presents a new DP algorithm for partition selection (a.k.a. set union), with an emphasis on parallelization. The proposed algorithm is provably better than the naive parallel baseline, and empirically outperforms a prior parallel algorithm. Its utility is weaker compared to earlier non-parallel algorithms. One nice aspect of their analysis is the idea of shifting weights without increasing sensitivity - this is somewhat reminiscent of the "waterfill" algorithm by Gopi et al., though that approach was sequential and didn't involve routing weights back to users. All the reviewers agree that this is an interesting paper that would make a solid contribution to ICML.